# ADVANTAGE WEIGHTED MATCHING: ALIGNING RL WITH PRETRAINING IN DIFFUSION MODELS

## ABSTRACT

Reinforcement Learning (RL) has emerged as a central paradigm for advancing Large Language Models (LLMs), where both pre-training and RL post-training stages are grounded in the same log-likelihood formulation. In contrast, recent RL approaches for diffusion models, most notably Denoising Diffusion Policy Optimization (DDPO), optimize an objective different from the pretraining objectives–score/flow matching loss. In this work, we establish a novel theoretical analysis: DDPO is an implicit form of score/flow matching with noisy targets, which increases variance and slows convergence. Building on this analysis, we introduce **Advantage Weighted Matching (AWM)**, a policy-gradient method for diffusion. It uses the same score/flow-matching loss as pretraining to obtain a lower-variance objective and reweights each sample by its advantage. In effect, AWM raises the influence of high-reward samples and suppresses low-reward ones while keeping the modeling objective identical to pretraining. This unifies pretraining and RL conceptually and practically, is consistent with policy-gradient theory, and reduces variance, yielding faster convergence. This simple yet effective design yields substantial benefits: on the GenEval, OCR, and PickScore benchmarks, AWM delivers up to a $24\times$ **speedup** over Flow-GRPO (which builds on DDPO), when applied to Stable Diffusion 3.5 Medium and FLUX, without compromising generation quality.

## 1 INTRODUCTION

Diffusion Models (Sohl-Dickstein et al., 2015; Ho et al., 2020; Song et al., 2020) and their variants, (Gaussian) Flow Matching (Peluchetti, 2023; Lipman et al., 2022; Albergo et al., 2023) are de facto generation paradigms for continuous domains like image and video synthesis (Nichol et al., 2021; Saharia et al., 2022; Ramesh et al., 2022; Rombach et al., 2022; Chen et al., 2023b; Brooks et al.,

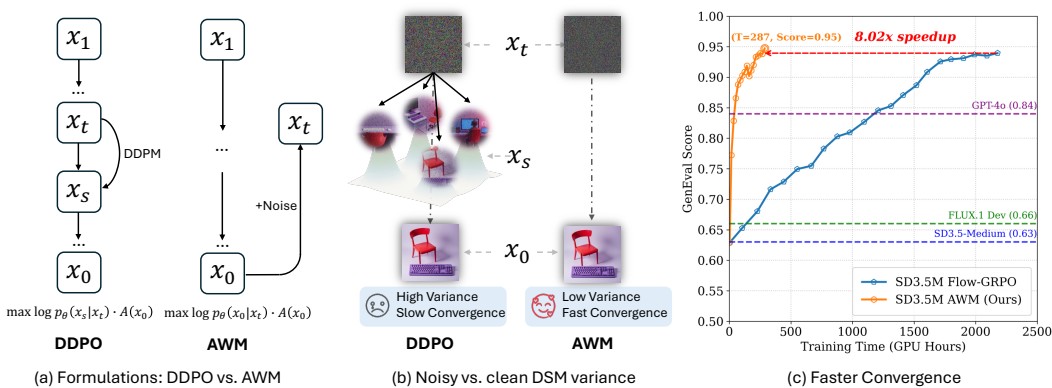

(a) Formulations: DDPO vs. AWM    (b) Noisy vs. clean DSM variance    (c) Faster Convergence

Figure 1: **AWM reduces variance and speeds up RL for diffusion.** (a) *Formulations:* DDPO optimizes per-step Gaussian likelihood on $x_{t-1}$, whereas AWM applies reward-weighted score/flow matching on $x_0$. (b) *Target variance:* DSM with noisy conditioning $\boldsymbol{x}_s$ (DDPO implicitly doing) has higher variance than DSM with clean $\boldsymbol{x}_0$ (AWM). (c) *Convergence:* On GenEval, AWM reaches the same quality with up to $8\times$ fewer GPU hours than Flow-GRPO.

2024). Motivated by the success of Reinforcement Learning (RL) for Large Language Models (LLMs) (Ziegler et al., 2019; Stiennon et al., 2020; Ouyang et al., 2022; Bai et al., 2022; Guo et al., 2025), there has been a growing interest in extending RL methodologies to diffusion models.

A key difference between RL algorithms used in LLMs and diffusion lies in the alignment of training objectives. In the autoregressive setting (*e.g.* LLMs), both pre-training and RL post-training ultimately optimize the same log-likelihood objective with different weights. In contrast, diffusion pretraining relies on score/flow matching losses that are closely related to maximum likelihood through the Evidence Lower Bound (ELBO) (Song et al., 2021) via forward process, while recent RL post-training methods for diffusion have largely adopted Denoising Diffusion Policy Optimization (DDPO) (Black et al., 2023), including Flow-GRPO (Liu et al., 2025) and Dance-GRPO (Xue et al., 2025). As shown in Figure 1.a (left), these approaches formulate denoising as a multi-step MDP, where each reverse-time step is an action, and the one-step reverse transition $p_{\boldsymbol{\theta}}(\boldsymbol{x}_{t-1}|\boldsymbol{x}_t, \boldsymbol{c})$ acts as the policy. Under standard DDPM samplers, this transition is isotropic Gaussian, enabling tractable per-step log-likelihoods and supporting policy-gradient methods, e.g., REINFORCE (Williams, 1992).

Though effective, DDPO-based RL post-training leads to an optimization objective that diverges from the score or flow matching objectives used during pretraining. The discrepancy raises fundamental questions: ***Why do diffusion RL post-training and pretraining employ different likelihood formulations? Consequently, what does the DDPO loss objective actually optimize?*** To answer this question, we establish a theoretical connection between DDPO and score matching in Section 2. As shown in Figure 1.b (left), DDPO is implicit doing denoising score matching (DSM) with noisy data. We prove that using noisy data increases the variance of the DSM target compared to the clean-data DSM objectives used in pretraining (shown in Figure 1.b (right)). These theoretical insights are validated in Section 2 through class-conditioned pretraining experiments on CIFAR-10 and ImageNet-64, where we show that noisy-DSM objectives align with the DDPO target but converge more slowly, consistent with our variance analysis.

We propose Advantage Weighted Matching (AWM), a simple yet effective objective that directly incorporates reward signals into score/flow matching without relying on reverse-time discretization. In contrast to DDPO, which depends on per-step likelihoods derived from reverse-time discretization, AWM uses the same score/flow-matching loss as diffusion pretraining while reweighting each sample by its advantage, as shown in Figure 1.a (right). Intuitively, AWM amplifies high-reward samples and downweights low-reward ones, while preserving the original pretraining objective. This design offers several benefits. First, it avoids the variance amplification caused by conditioning on noisy data, leading to faster convergence. Second, it decouples training from sampling: AWM supports any sampler or noise level, unlike DDPO methods tied to Euler–Maruyama discretization. Finally, by grounding RL post-training in the same score/flow matching framework as pretraining, AWM restores conceptual symmetry between diffusion models and LLMs, where both stages optimize the same objective with reward-dependent weighting.

AWM translates this variance reduction into substantial wall-clock savings without compromising quality. On Stable Diffusion 3.5 Medium (SD3.5M) (Esser et al., 2024), it matches Flow-GRPO's (Liu et al., 2025) GenEval score (0.95 vs. 0.95) with an 8.02× speed-up (Figure 1.c). It achieves comparable OCR accuracy with 23.6× less compute and the PickScore with 10.5× less. On FLUX (Labs, 2024), AWM shows 8.5× (OCR) and 6.8× (PickScore) speedups (Figure 5 a–d), reaching 0.986 OCR accuracy. Our contributions are threefold:

- **DDPO–DSM Equivalence.** We prove that, up to discretization error, maximizing DDPO's per-step Gaussian likelihood is equivalent to minimizing denoising score matching (DSM) with *noisy data* at the same time step, regardless of parameterization (score or velocity).
- **Additional Variance for DDPO from Noisy Conditioning.** We show that conditioning on noisy data yields a higher-variance estimator of score function than conditioning on clean data; we quantify the increase and validate it via controlled pretraining on CIFAR-10 and ImageNet-64, where noisy-DSM converges more slowly under identical settings.
- **Advantage Weighted Matching (AWM).** We introduce Advantage Weighted Matching, a policy gradient method for diffusion with forward process that decouples training from sampling, and realigns RL post-training with the pretraining objective. On GenEval, OCR, and PickScore, AWM achieves up to $24\times$ faster training than Flow-GRPO (Liu et al., 2025) on SD-3.5-M and FLUX without generation quality degradation.

## 2 DDPO Is Secretly Doing Score Matching

### 2.1 Background on Diffusion and Flow Matching

Diffusion Models (Sohl-Dickstein et al., 2015; Ho et al., 2020; Song et al., 2020) diffuse clean data sample $x_0 \sim p_{\text{data}}$ from data distribution to noisy data $x_t = \alpha_t x_0 + \sigma_t \epsilon$, where $t \in [0, T]$, $\epsilon \sim \mathcal{N}(0, I)$ is a standard Gaussian noise. $x_T$ exactly or approximately follows a Gaussian distribution. Typically, diffusion models train a noise prediction network $\epsilon_\theta$ using $\mathbb{E}_{x_0, \epsilon, t}[\|\epsilon_\theta(x_t, t) - \epsilon\|^2]$, which is equivalent to denoising score matching loss (Vincent, 2011; Song et al., 2020).

Flow Matching (Peluchetti, 2023; Liu et al., 2022; Lipman et al., 2022; Albergo et al., 2023) with Gaussian priors considers a linear interpolation noising process by defining $x_t = (1 - t)x_0 + t\epsilon$. The flow matching models train a velocity prediction network $v_\theta$ using $\mathbb{E}_{x_0, \epsilon, t}[\|v_\theta(x_t, t) - (\epsilon - x_0)\|^2]$. To simplify the notation, in the remaining part of the paper, we choose the flow matching noise schedule by default.

The sampling process of diffusion or flow matching models involves solving the probability flow ODE (PF-ODE) (Song et al., 2020): $dx_t = v_\theta(x_t, t)dt$, or diffusion SDE (Song et al., 2020), where $w_t$ represents the Wiener process[1]:

$$dx_t = \left[ v_\theta(x_t, t) + \frac{1}{1-t}(x_t + (1-t)v_\theta(x_t, t)) \right]dt + \sqrt{\frac{2t}{1-t}}dw_t, \tag{1}$$

The likelihood $\log p(x)$ of data $x$ that diffusion models assign can be exactly computed through instantaneous change of variables as in Neural ODE (Chen et al., 2018) or approximated by an Evidence Lower Bound (ELBO) (Song et al., 2021; Kingma et al., 2021; Kingma & Gao, 2023):

$$-\log p(x) \le \text{ELBO}(x) := \mathbb{E}_{t \sim \mathcal{U}(0,1), \epsilon \sim \mathcal{N}(0,I)}\left[ \frac{1-t}{t}\|v_\theta(x_t, t) - (\epsilon - x_0)\|^2 \right]. \tag{2}$$

The gap between the exact likelihood and ELBO is relatively small (Song et al., 2021).

### 2.2 Denoising Diffusion Policy Optimization (DDPO)

Reinforcement learning (RL) algorithms in the context of Large Language Models (LLMs) vary in their data utilization strategies (online, offline, or mixed) and in the algorithms used, *e.g.* preference-based methods like DPO (Clark et al., 2023), and policy gradient variants (importance sampling, clipping, with or without a value-function baseline) such as PPO (Schulman et al., 2017) and GRPO (Shao et al., 2024). Despite these differences, the likelihood formulation of the policy model $\pi_\theta$ remains consistent, since autoregressive model allows exact computation of sequence likelihoods.

Pretraining and RL post-training in LLMs are conceptually aligned. Pretraining maximizes the teacher-forced sequence log-likelihood, while RL applies reward-weighted adjustments to the same likelihood—amplifying high-reward samples and downweighting low-reward ones (e.g., via advantages or preference scores). In both cases, the objective centers on the model's sequence log-likelihood; only the weighting differs.

In contrast, RL for diffusion models, e.g., Flow-GRPO (Liu et al., 2025) and Dance-GRPO (Xue et al., 2025), adopts the Denoising Diffusion Policy Optimization (DDPO) framework (Black et al., 2023). DDPO casts denoising as a multi-step MDP: the state at step $t$ is $x_t$, the policy is the one-step reverse transition $p_\theta(x_{t-1}|x_t)$. Under standard DDPM samplers (or an Euler–Maruyama discretization of the reverse-time diffusion SDE), this transition is isotropic Gaussian, which makes the per-step conditional log-likelihood tractable and fit for policy-gradient style optimization. Concretely, DDPO implements the Euler–Maruyama update of the reverse SDE from Equation (1)[2]:

$$x_{t-\Delta t} = x_t - \left[ v_\theta(x_t, t) + \frac{1}{1-t}(x_t + (1-t)v_\theta(x_t, t)) \right]\Delta t + \sqrt{\frac{2t}{1-t}}\sqrt{\Delta t}\epsilon, \tag{3}$$

---

[1]Or an extension with arbitrary noise level (Karras et al., 2022; Xue et al., 2023)

[2]For simplicity, our analysis uses the vanilla reverse-time diffusion SDE. Euler–Maruyama can increase estimator variance, and using an exponential integrator yields the analytical variance, as discussed in (Xue et al., 2023). The original DDPO (Black et al., 2023) uses DDPM, which can be viewed as Euler–Maruyama with an exponential integrator (Xue et al., 2023). Flow-GRPO (Liu et al., 2025) and Dance-GRPO (Xue et al., 2025) use the plain Euler–Maruyama discretization. The discrepancy vanishes as the discretization step $\Delta t \to 0$.

For completeness, we include a detailed description of DDPO in Appendix D. This implies a Gaussian policy $p_{\boldsymbol{\theta}}(\boldsymbol{x}_{t-\Delta t}|\boldsymbol{x}_t)$ with tractable mean and covariance, yielding the per-step log-likelihood $\log p_{\boldsymbol{\theta}}(\boldsymbol{x}_{t-\Delta t}|\boldsymbol{x}_t)$:

$$-\left\|\boldsymbol{x}_{t-\Delta t} - \left(\boldsymbol{x}_t - \left[\boldsymbol{v}_{\boldsymbol{\theta}}(\boldsymbol{x}_t, t) + \frac{1}{1-t}\left(\boldsymbol{x}_t + (1-t)\boldsymbol{v}_{\boldsymbol{\theta}}(\boldsymbol{x}_t, t)\right)\right]\Delta t\right)\right\|^2 / (\frac{4t}{1-t}\Delta t) + \text{const.} \tag{4}$$

By contrast, standard pretraining for diffusion optimizes a score/flow-matching objective:

$$\|\boldsymbol{v}_{\boldsymbol{\theta}}(\boldsymbol{x}_t, t) - (\boldsymbol{\epsilon} - \boldsymbol{x}_0)\|^2. \tag{5}$$

While score/flow matching is itself a likelihood surrogate for diffusion pretraining, DDPO optimizes a per-step reverse-transition likelihood: one works on the forward process with clean conditioning, the other on a discretized reverse process with noisy intermediates. This raises the pivot question: *Why do reinforcement learning and pretraining for diffusion models employ different likelihood formulations? Consequently, what is the DDPO loss objective actually optimizing?*

### 2.3 DENOISING SCORE MATCHING WITH NOISY DATA

This subsection proceeds as in Fig. 2: (i) Theorem 1—a DDPO step equals DSM on $\boldsymbol{x}_t$ conditioned on noisy $\boldsymbol{x}_{t-\Delta t}$; (ii) Lemma 1—noisy-DSM and clean-DSM share the same population minimizer; (iii) Theorem 2—noisy conditioning strictly increases target variance, explaining slower optimization. We validate (iii) with EDM experiments (Figure 3).

Leveraging the fact *the joint distribution of $(\boldsymbol{x}_{t-\Delta t}, \boldsymbol{x}_t)$ is the same for forward and reverse process* (Haussmann & Pardoux, 1986) and omitting Euler–Maruyama discretization error, our main result shows that a DDPO step at time $t$ targets the same population objective as denoising score matching (DSM) conditioned on noisy $\boldsymbol{x}_{t-\Delta t}$.

**Theorem 1** (DDPO is Secretly Doing Denoising Score Matching with Noisy Data). *Optimizing the DDPO objective Equation (4), which maximizes the log-likelihood derived from the discretization of the reverse-time diffusion SDE, is equivalent to minimizing the Denoising Score Matching loss with noisy data from the forward process, omitting the discretization error induced by Euler-Maruyama discretization:*

$$\mathbb{E}_{\boldsymbol{x}_{t-\Delta t}, \boldsymbol{x}_t}\left[\|\boldsymbol{s}_{\boldsymbol{\theta}}(\boldsymbol{x}_t, t) - \nabla \log p(\boldsymbol{x}_t|\boldsymbol{x}_{t-\Delta t})\|^2\right]. \tag{6}$$

The proof can be found in Appendix C. Theorem 1 establishes that maximizing DDPO's per-step log-likelihood aligns with minimizing DSM with noisy data. This equivalence holds for both score and velocity parameterizations. In the following lemma and theorem, we will show that optimizing Denoising Score Matching with noisy data is equivalent to optimizing Denoising Score Matching with clean data in expectation. However, ***conditioning on a noisy $\boldsymbol{x}_{t-\Delta t}$ introduces additional variance in the DSM target***. This justifies analyzing DDPO through the lens of score matching while isolating variance effects that impact optimization.

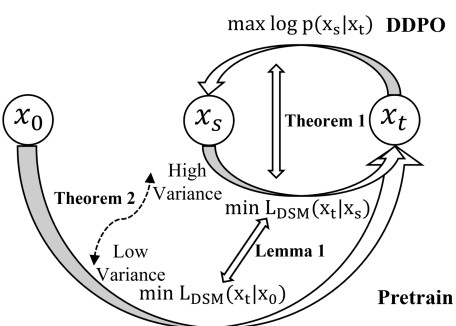

Figure 2: Overview of our analysis.

**Lemma 1** (Denoising Score Matching with Noisy Data). *For any time steps $s$ and $t$ such that $0 \le s < t$, optimizing the Denoising Score Matching objective using the noisy data $\boldsymbol{x}_s$*

$$\mathbb{E}_{\boldsymbol{x}_s}\mathbb{E}_{\boldsymbol{x}_t|\boldsymbol{x}_s}\left[\|\boldsymbol{s}_{\boldsymbol{\theta}}(\boldsymbol{x}_t, t) - \nabla \log p(\boldsymbol{x}_t|\boldsymbol{x}_s)\|^2\right],$$

*is equivalent to optimizing the standard Score Matching objective:*

$$\mathbb{E}_{\boldsymbol{x}_t}\left[\|\boldsymbol{s}_{\boldsymbol{\theta}}(\boldsymbol{x}_t, t) - \nabla \log p(\boldsymbol{x}_t)\|^2\right].$$

*This lemma generalizes the well-known result that Denoising Score Matching with clean data is equivalent to Score Matching (Vincent, 2011; Tang & Zhao, 2025), which corresponds to the special case in this lemma when $s = 0$.*

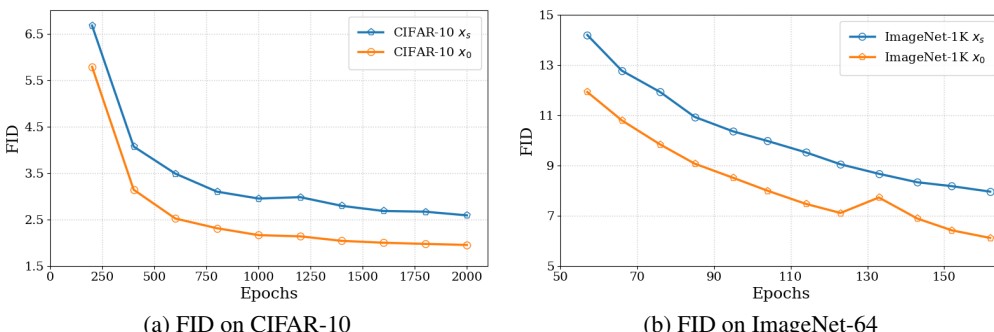

(a) FID on CIFAR-10                  (b) FID on ImageNet-64

Figure 3: **Noisy conditioning increases variance and hinders pretraining.** We compare the EDM baseline ($x_0$, clean-data objective; Equation (7)) with a noisy-conditioned proxy ($x_s$; Equation (8)) aligned with DDPO. Under the same constraints, the noisy objective yields consistently worse FID on CIFAR-10 and ImageNet-64—consistent with Theorem 2.

The proof is in Appendix C. DDPO update aligns with a DSM objective at the same $t$ (up to discretization error), so both share the same population minimizer. What remains is to quantify the optimization noise. The next theorem shows that noisy conditioning inflates DSM target variance.

**Theorem 2** (Variance of Denoising Score Matching Target with Noisy Data is Larger). *For the two conditional scores*

$$\nabla_{\boldsymbol{x}_t} \log p(\boldsymbol{x}_t|\boldsymbol{x}_s), \qquad \nabla_{\boldsymbol{x}_t} \log p(\boldsymbol{x}_t|\boldsymbol{x}_0),$$

*and for any $\boldsymbol{x}_t$ and any $s \in [0, t)$, each is unbiased estimator of $\nabla_{\boldsymbol{x}_t} \log p(\boldsymbol{x}_t)$:*

$$\mathbb{E}\left[\nabla_{\boldsymbol{x}_t} \log p(\boldsymbol{x}_t|\boldsymbol{x}_s)|\boldsymbol{x}_t\right] = \mathbb{E}\left[\nabla_{\boldsymbol{x}_t} \log p(\boldsymbol{x}_t|\boldsymbol{x}_0)|\boldsymbol{x}_t\right] = \nabla_{\boldsymbol{x}_t} \log p(\boldsymbol{x}_t),$$

*and their conditional covariances satisfy*

$$\mathrm{Cov}\left(\nabla_{\boldsymbol{x}_t} \log p(\boldsymbol{x}_t|\boldsymbol{x}_s)|\boldsymbol{x}_t\right) = \mathrm{Cov}\left(\nabla_{\boldsymbol{x}_t} \log p(\boldsymbol{x}_t|\boldsymbol{x}_0)|\boldsymbol{x}_t\right) + \kappa(s,t)\,I \succeq \mathrm{Cov}\left(\nabla_{\boldsymbol{x}_t} \log p(\boldsymbol{x}_t|\boldsymbol{x}_0)|\boldsymbol{x}_t\right),$$

*where*

$$\kappa(s,t) = \frac{(1-t)^2\,s^2}{t^2\left(t^2(1-s)^2 - s^2(1-t)^2\right)}.$$

*In particular,* $\mathrm{Tr}\,\mathrm{Cov}(\nabla_{\boldsymbol{x}_t} \log p(\boldsymbol{x}_t|\boldsymbol{x}_s)|\boldsymbol{x}_t) = \mathrm{Tr}\,\mathrm{Cov}(\nabla_{\boldsymbol{x}_t} \log p(\boldsymbol{x}_t|\boldsymbol{x}_0)|\boldsymbol{x}_t) + d\,\kappa(s,t)$, *where $d$ is the data dimension. The map $s \mapsto \kappa(s,t)$ is strictly increasing on $[0,t)$, with $\kappa(0,t) = 0$.*

The proof appears in Appendix C. The theorem establishes that, although noisy-conditioned DSM is unbiased and targets the same minimizer as clean-conditioned DSM, its target has strictly larger conditional covariance (by $d\,\kappa(s,t)$), which increases the objective variance and slows stochastic optimization (Wang et al., 2013). We corroborate this prediction with pretraining experiments on CIFAR-10 and ImageNet-64 using the EDM (Karras et al., 2022)[3] codebase. Our baseline uses the standard clean-data objective Equation (7), where $\boldsymbol{D_\theta}$ denotes the data prediction network:

$$\mathbb{E}_{\boldsymbol{x}_0,\boldsymbol{x}_t}\left[w(t)\,\|\boldsymbol{D_\theta}(\boldsymbol{x}_t,t) - \boldsymbol{x}_0\|^2\right] \tag{7}$$

Under the $\boldsymbol{x}_0$-prediction framework (consistent with EDM), the noisy Denoising Score Matching (DSM) objective is formulated as Equation (8), which mirrors the DDPO objective. The detailed derivation is provided in the Appendix C.4:

$$\mathbb{E}_{\boldsymbol{x}_s,\boldsymbol{x}_t}\left[w(t)\,\left\|\boldsymbol{D_\theta}(\boldsymbol{x}_t,t) - \left(\boldsymbol{x}_t - \frac{t^2}{t^2 - s^2}(\boldsymbol{x}_t - \boldsymbol{x}_s)\right)\right\|^2\right] \tag{8}$$

Under identical architectures, noise schedules, and optimization settings, the noisy-DSM objective learns in the correct direction but reaches a given quality level substantially later (see Figure 3), consistent with the variance analysis.

## 3 ADVANTAGE WEIGHTED MATCHING

Motivated by our finding that DDPO implicitly optimizes DSM with *noisy* data, which increases variance, we introduce Advantage Weighted Matching (AWM), replacing the DDPO objective with

---

[3]EDM's noise scheduler assumes that $\boldsymbol{x}_t = \boldsymbol{x}_0 + t\epsilon$, $t \in [0.002, 80]$ during inference and $t \in [0, +\infty]$ during training.

Figure 4: **AWM pipeline.** For each prompt $\boldsymbol{c}$, we sample a group of sequences $\{\boldsymbol{x}_0\}$ from a diffusion model, evaluate rewards, advantages, and train the sequence policy via advantage-weighted score matching. During inference, any ODE/SDE sampler can be used.

DSM evaluated on *clean* data. Conceptually, this converts a step-wise reverse-transition policy into a sequence-level conditional policy over $\boldsymbol{x}_0$. The resulting difference in problem setup is:

**Problem Setup of Denoising Diffusion Policy Optimization**

State: $\quad \boldsymbol{s}_t = (\boldsymbol{c}, t, \boldsymbol{x}_t) \qquad$ Action: $\quad \boldsymbol{a}_t = \boldsymbol{x}_{t-1} \qquad$ Policy: $\quad \pi(\boldsymbol{a}_t|\boldsymbol{s}_t) = p_{\boldsymbol{\theta}}(\boldsymbol{x}_{t-1}|\boldsymbol{x}_t, \boldsymbol{c})$

**Problem Setup of Advantage Weighted Matching**

State: $\quad \boldsymbol{s} = \boldsymbol{c} \qquad\qquad$ Action: $\quad \boldsymbol{a} = \boldsymbol{x}_0 \qquad\quad$ Policy: $\quad \pi(\boldsymbol{a}|\boldsymbol{s}) = p_{\boldsymbol{\theta}}(\boldsymbol{x}_0|\boldsymbol{c})$

As show in Figure 4, for a prompt $\boldsymbol{c}$ and a batch $\{\boldsymbol{x}_i\}_{i=1}^{G}$ sampled from distribution $\pi_{\boldsymbol{\theta}_{\mathrm{old}}}$, we compute rewards $r_i = r(\boldsymbol{x}_i, \boldsymbol{c})$ and advantages $A_i$ (e.g., through group-relative mean $A_i = \frac{r_i - \mathrm{mean}(r_i)}{\mathrm{std}(r_i)}$, where $\mathrm{std}(r_i)$ denotes the global std across all reward value in this batch or local std within the group. We adopt the former global one). The finally objective for the Advantage Weighted Matching under the GRPO framework is as follows, where $\pi_{\boldsymbol{\theta}_{\mathrm{old}}}$ is the policy used to collect the sampling data, used for computing importance sampling ratios in GRPO/PPO, and $\pi_{\boldsymbol{\theta}_{\mathrm{ref}}}$ is the frozen reference model (usually the pretrained base model), used to calculate the KL divergence penalty to prevent reward hacking:

$$\mathcal{J}_{GRPO}(\boldsymbol{\theta}) = \mathbb{E}_{\boldsymbol{c}:\ \mathrm{prompt}, \{\boldsymbol{x}_i\}_{i=1}^{G} \sim \pi_{\boldsymbol{\theta}_{\mathrm{old}}}(\boldsymbol{x}|\boldsymbol{c})} \frac{1}{G} \sum_{i=1}^{G} \left( \frac{\pi_{\boldsymbol{\theta}}(\boldsymbol{x}_i|\boldsymbol{c})}{\pi_{\boldsymbol{\theta}_{\mathrm{old}}}(\boldsymbol{x}_i|\boldsymbol{c})} \cdot A_i - \beta \mathbb{D}_{KL}(\pi_{\boldsymbol{\theta}} \| \pi_{\boldsymbol{\theta}_{\mathrm{ref}}}) \right). \quad (9)$$

For simplicity, we omit the clip operation here. The policy likelihood can be substituted by score/flow matching loss, which is a tight ELBO of likelihood $p_{\boldsymbol{\theta}}(\boldsymbol{x}_0|\boldsymbol{c})$ (Song et al., 2021; Kingma et al., 2021; Kingma & Gao, 2023). We view the sequence policy $\log \pi_\theta(\boldsymbol{x}_0|\boldsymbol{c})$ through its ELBO surrogate given by score/flow matching (Song et al., 2021; Kingma et al., 2021):

$$-\mathbb{E}_t \left[ w(t) \left\| \boldsymbol{v}_{\boldsymbol{\theta}}(\boldsymbol{x}_t, t, \boldsymbol{c}) - (\boldsymbol{\epsilon} - \boldsymbol{x}_0) \right\|^2 \right]. \quad (10)$$

where $w(t)$ is the standard time-weight (e.g., ELBO). In practice, we find that the uniform weight $w(t) = 1$ works better. Such a behavior roots in the different nature of the visual quality and likelihood for diffusion models and has been observed in (Ho et al., 2020; Nichol & Dhariwal, 2021; Song et al., 2021; Vahdat et al., 2021; Watson et al., 2021; Kingma et al., 2021). Thus, the likelihood ratio $\frac{\pi_{\boldsymbol{\theta}}(\boldsymbol{x}|\boldsymbol{c})}{\pi_{\boldsymbol{\theta}_{\mathrm{old}}}(\boldsymbol{x}|\boldsymbol{c})}$ can be estimated through:

$$\exp \left( -\mathbb{E}_t \left[ w(t) \left\| \boldsymbol{v}_{\boldsymbol{\theta}}(\boldsymbol{x}_t, t, \boldsymbol{c}) - (\boldsymbol{\epsilon} - \boldsymbol{x}_0) \right\|^2 - w(t) \left\| \boldsymbol{v}_{\boldsymbol{\theta}_{\mathrm{old}}}(\boldsymbol{x}_t, t, \boldsymbol{c}) - (\boldsymbol{\epsilon} - \boldsymbol{x}_0) \right\|^2 \right] \right), \quad (11)$$

for which we use the shared timesteps and noise for $\pi_{\boldsymbol{\theta}}$ and $\pi_{\boldsymbol{\theta}_{\mathrm{old}}}$ from LLaDA 1.5 (Zhu et al., 2025) for variance reduction. The KL term can be estimated (Song et al., 2021) through:

$$w(t) \left\| \boldsymbol{v}_{\boldsymbol{\theta}}(\boldsymbol{x}_t, t, \boldsymbol{c}) - \boldsymbol{v}_{\boldsymbol{\theta}_{\mathrm{ref}}}(\boldsymbol{x}_t, t, \boldsymbol{c}) \right\|^2. \quad (12)$$

To make the mechanics explicit, consider an *on-policy* update on a single sample $\boldsymbol{x}_i$ drawn from $\pi_\theta(\cdot \mid \boldsymbol{c})$. Let $\hat{\pi}_\theta$ denote the ELBO-based surrogate of the sequence likelihood,

$$\nabla_{\boldsymbol{\theta}} \frac{\hat{\pi}_{\boldsymbol{\theta}}(\boldsymbol{x}_i|\boldsymbol{c})}{\mathtt{stopgrad}\left(\hat{\pi}_{\boldsymbol{\theta}}(\boldsymbol{x}_i|\boldsymbol{c})\right)} \cdot A_i = \nabla_{\boldsymbol{\theta}} \log \hat{\pi}_{\boldsymbol{\theta}}(\boldsymbol{x}_i|\boldsymbol{c}) \cdot A_i = -\nabla_{\boldsymbol{\theta}} \mathbb{E}_t \left[ w(t) \left\| \boldsymbol{v}_{\boldsymbol{\theta}}(\boldsymbol{x}_t, t, \boldsymbol{c}) - (\boldsymbol{\epsilon} - \boldsymbol{x}_0) \right\|^2 \right] \cdot A_i.$$

---

**Algorithm 1:** Pseudo code of the training loop for Advantage Weighted Matching

```
1  for i in range(num_training_steps):
2      samples = sampler(model, prompt) # get samples with arbitrary sampler
3      reward = reward_fn(samples) # get reward of samples
4      advantage = cal_adv(reward, prompt) # e.g. group relative mean
5
6      noise = randn_like(samples)
7      timesteps = get_timesteps(samples)
8      noisy_samples = fwd_diffusion(samples, noise, timesteps) # add noise
9      velocity_pred = model(noisy_samples, timesteps, prompt)
10     velocity_ref = ref_model(noisy_samples, timesteps, prompt) # optional for kl loss
11
12     log_p = -((velocity_pred - (noise-samples))**2).mean() # Flow Matching Loss
13     ratio = torch.exp(log_p - log_p.detach()) # or log_p_old for off-policy update
14     policy_loss = -advantage * ratio
15     kl_loss = weight(timesteps)*((velocity_pred - velocity_ref)**2).mean()
16     loss = policy_loss + beta * kl_loss
```

---

When $A_i > 0$ (good sample), the gradient decreases the flow matching loss at $(\boldsymbol{x}_i, \boldsymbol{c})$, pulling $\boldsymbol{v}_\theta$ toward the target $(\boldsymbol{\epsilon} - \boldsymbol{x}_i)$; when $A_i < 0$ (bad sample), the gradient will push the velocity away from the undesirable target.

We provide a clear pseudo-code and pipeline Figure 4 for our AWM algorithm. For each prompt $\boldsymbol{c}$, we (i) sample a group of samples $\{\boldsymbol{x}_0\}_{i=1}^{G}$, (ii) evaluate rewards and form group-relative advantages $A_i$, (iii) get noisy data $\boldsymbol{x}_t = (1-t)\boldsymbol{x}_0 + t\boldsymbol{\epsilon}$ to evaluate score matching losses, and (iv) optimize an *advantage-weighted* FM objective with a velocity-space KL regularization term. Concretely:

We summarize several key differences between AWM and DDPO below:

**Variance of Policy Gradient**  Subtracting a baseline in policy gradient will keep the gradient unbiased and reduce the variance as an additive control variate (Greensmith et al., 2004). For diffusion models, there is another level of variance that comes from the estimation of the likelihood as the exact likelihood is computationally intractable. AWM reduces the variance compared to DDPO.

**Decouple of Sampling and Training**  The forward process formulation decouples the training and sampling procedure of AWM. This has several advantages: First, the samplers are not restricted to DDPM or Euler-Maruyama; we can use more advanced ODE samplers (Lu et al., 2022) or SDE samplers (Xue et al., 2023) for a better trade-off between quality and speed, which we leave for future work. Second, the training timesteps and sampling timesteps can be decoupled, *e.g.*, we can use 20 steps for sampling and 4 steps for training. Third, this also allows future explorations, such as using the step-distilled model to accelerate sampling.

**Alignment with Pretraining**  AWM keeps the conceptual symmetry between pretraining and RL for diffusion as in LLM: both optimize the same DSM/FM loss, differing only by *reward-derived weights*. Also, this avoids the use of CFG (Ho & Salimans, 2022) in training as pretraining.

## 4 EXPERIMENTS

We use two representative open-sourced models: SD3.5M (Esser et al., 2024) and Flux (Labs, 2024) on the above three reward tasks, i.e., composition image generation on GenEval (Ghosh et al., 2023), visual-text rendering on OCR (Chen et al., 2023a), and human preference alignment on PickScore(Kirstain et al., 2023). We use a group size $G = 24$, and LoRA (Hu et al., 2022) with $\alpha = 64$ and $r = 32$ for SD3.5M and LoRA with $\alpha = 128$ and $r = 64$ for FLUX. The KL ratio $\beta$ is set to $0.4$ for GenEval and OCR, and $0.01$ for PickScore. The learning rate is set to a constant $3e-4$. We keep $w(t) = 1$ and Euler-Maruyama as our default sampler throughout our experiments.

### 4.1 MAIN RESULTS

Table 1 reports the comparative performance of various models on the GenEval benchmark for SD3.5M. Among popular models, e.g., SD3.5L, DALLE-3, GPT-4o, and FLUX.1 Dev, overall

Table 1: Performance comparison on GenEval. Speed-up is relative to Flow-GRPO baseline.

| Model | Single Obj. | Two Obj. | Counting | Color | Position | Attr | Overall | Speed-up |
|---|---|---|---|---|---|---|---|---|
| | \multicolumn{8}{c}{Geneval} | | | | | | | |
| DALLE-3 | 0.96 | 0.87 | 0.47 | 0.83 | 0.43 | 0.45 | 0.67 | – |
| GPT-4o | 0.99 | 0.92 | 0.85 | 0.92 | 0.75 | 0.61 | 0.84 | – |
| SD-XL | 0.98 | 0.74 | 0.39 | 0.85 | 0.15 | 0.23 | 0.55 | – |
| FLUX.1 Dev | 0.98 | 0.81 | 0.74 | 0.79 | 0.22 | 0.45 | 0.66 | – |
| SD3.5L | 0.98 | 0.89 | 0.73 | 0.83 | 0.34 | 0.47 | 0.71 | – |
| SD3.5M | 0.98 | 0.78 | 0.50 | 0.81 | 0.24 | 0.52 | 0.63 | – |
| Flow-GRPO | 1.00 | 0.99 | 0.95 | 0.92 | 0.99 | 0.86 | **0.95** | $1\times$ |
| AWM (Ours) | 1.00 | 0.99 | 0.95 | 0.93 | 0.98 | 0.83 | **0.95** | $\mathbf{8.02\times}$ |

Table 2: Performance comparison on OCR/PickScore for SD3.5M and FLUX. Hours is the total GPU hours; Speed-up is relative to Flow-GRPO baseline. [†] indicates longer training time.

| Method | \multicolumn{4}{c}{SD3.5M} | | | | \multicolumn{4}{c}{FLUX} | | | |
| | \multicolumn{2}{c}{OCR} | | \multicolumn{2}{c}{PickScore} | | \multicolumn{2}{c}{OCR} | | \multicolumn{2}{c}{PickScore} | |
| | Acc | Hours | Score | Hours | Acc | Hours | Score | GPU hours |
|---|---|---|---|---|---|---|---|---|
| Base model | 0.59 | – | 21.72 | – | 0.59 | – | 22.20 | – |
| FlowGRPO | 0.89 | 415.9 ($1\times$) | 23.01 | 956.1 ($1\times$) | 0.95 | 343.6 ($1\times$) | 23.08 | 339.2 ($1\times$) |
| AWM (Ours) | 0.89 | 17.6 ($23.59\times$) | 23.02 | 91.1 ($10.49\times$) | 0.95 | 40.3 ($8.53\times$) | 23.08 | 49.8 ($6.82\times$) |
| AWM (Ours)[†] | 0.95 (+6.74%) | 79.0 | 23.25 (+0.99%) | 205.0 | 0.99 (+4.21%) | 147.0 | 23.18 (+0.43%) | 78.0 |

scores range between 0.55 and 0.84, with GPT-4o achieving the highest performance (0.84). After RL post-training, Flow-GRPO reaches an overall GenEval score of 0.95, serving as the baseline, while AWM matches this score (0.95) but does so with a dramatic efficiency gain—achieving an 8.02× speed-up in GPU hours to convergence. Notably, AWM also attains consistently strong results across all sub-tasks, including Two-Object (0.99), Color (0.93), confirming that it preserves quality while substantially reducing training cost.

Besides, we further validated AWM on other metrics such as OCR and PickScore, and across different backbones, including SD3.5M and FLUX. As shown in Table 2, AWM surpasses Flow-GRPO in OCR and PickScore while requiring far fewer GPU hours. For example, SD3.5M with AWM achieves OCR (0.89) with a 23.6× speed-up, and OCR (0.95) with an 8.53× reduction on FLUX.

We illustrate training efficiency in Figure 5. AWM achieves comparable or superior performance on both OCR and PickScore while using significantly fewer GPU hours—for example, up to 23.6× faster on SD3.5M OCR and 6.8× faster on FLUX PickScore. These results demonstrate consistent efficiency gains over Flow GRPO across models and evaluation metrics. Specifically, AWM reaches an OCR score of 0.89 on SD3.5 using only 17.6 GPU hours, whereas Flow GRPO requires over 400 GPU hours to attain the same performance.

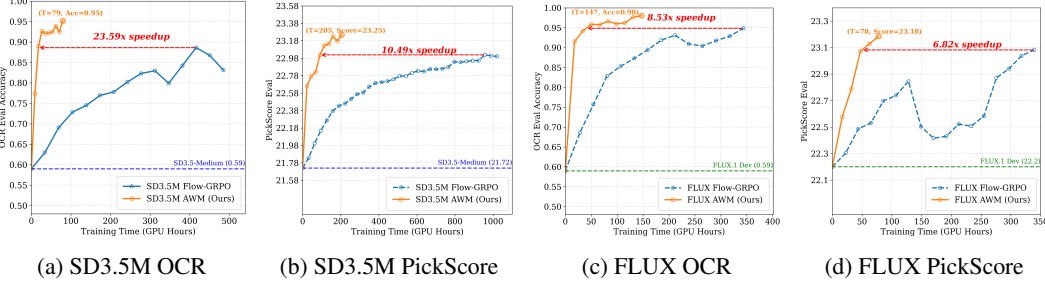

| (a) SD3.5M OCR | (b) SD3.5M PickScore | (c) FLUX OCR | (d) FLUX PickScore |

Figure 5: **OCR and PickScore training efficiency.** Metric vs. GPU hours for SD3.5M and FLUX. AWM (ours) exceeds Flow-GRPO with far less compute.

We visualize the FLUX baseline and FLUX after 100 AWM gradient steps on three composition prompts and two text-rendering (OCR) prompts. The composition prompts specify numerosity, color, and position constraints. As shown in the second column of Figure 6, AWM adheres closely to the instructions—e.g., "three," "purple," "fox," "one," "blue"—and likewise improves text rendering.

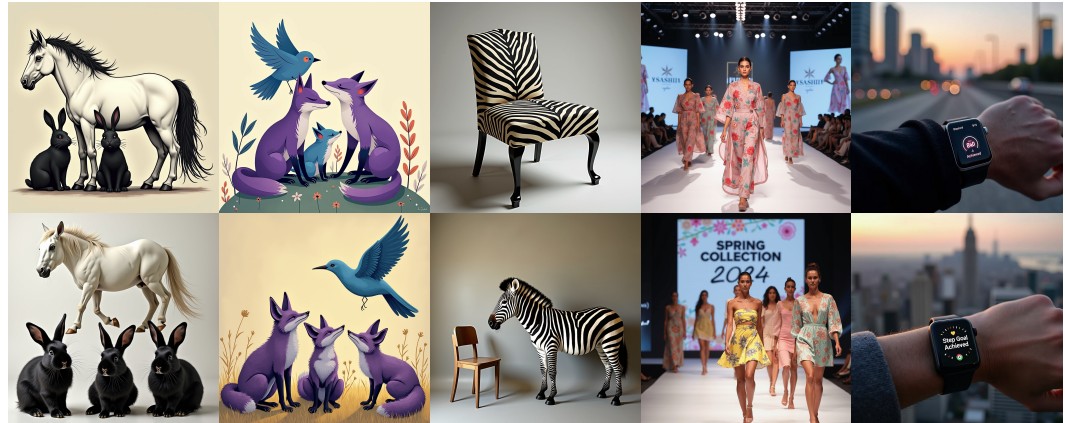

Figure 6: Visual comparison before (first row) and after 100 steps of AWM training (second row). Prompts from the GenEVAL and OCR benchmarks (listed in [4]) are used for generation.

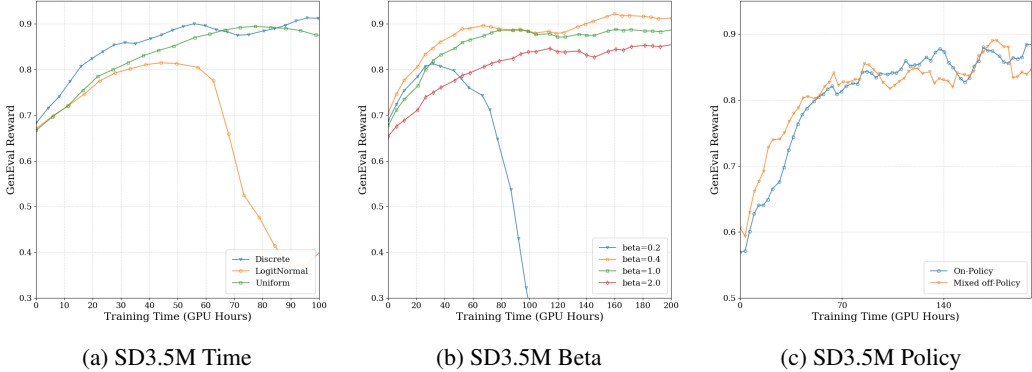

| (a) SD3.5M Time | (b) SD3.5M Beta | (c) SD3.5M Policy |

Figure 7: **Ablations on SD3.5M and GenEval** (a) Timestep sampling $p(t)$; (b) KL strength $\beta$; (c) Data reuse (on-policy vs. one-step off-policy). Curves show GenEval reward vs. GPU hours.

## 4.2 ABLATION STUDIES

Unless otherwise stated, all ablations are conducted on SD3.5M with identical architectures, batch sizes, and reward pipelines; curves report GenEval reward versus wall-clock (GPU hours). Results are summarized in Figure 7(a–c).

**Sampling distribution over timesteps $p(t)$.** We compare three choices: (i) a *discrete* distribution supported on the inference sampler's time grid $\{t_k\}$, (ii) *uniform* $t \sim \mathcal{U}(0, 1)$, and (iii) a *logit-normal* distribution that concentrates mass near mid-range $t$. Figure 7a shows that discrete and uniform achieve similar results, while logit-normal lags and eventually degrade during RL fine-tuning. We adopt discrete as our default settings.

**KL regularization strength $\beta$.** We sweep $\beta \in \{0.2, 0.4, 1.0, 2.0\}$ in the velocity-space KL proxy. As shown in Figure 7b, too little regularization ($\beta{=}0.2$) can lead to instability and collapse, while too much ($\beta{=}2.0$) slows learning and underperforms. A medium range ($\beta{\in}[0.4, 1.0]$) is stable and fast. Thus, the KL should stabilize the update, not dominate it.

**On-policy vs. mixed (one-step) off-policy.** We compare purely on-policy updates with a simple mixed strategy that reuses 50% of batches from the previous policy (one-step off-policy) with importance ratios. Figure 7c shows very similar learning curves and final scores; the mixed regime enables future extensions to deeper off-policy reuse, which we adopt as default settings.

---

[4]Prompts in Figure 6 are: (1) three black rabbits below one white horse; (2) three purple foxes below one blue bird; (3) a photo of a chair left of a zebra; (4) A high-fashion runway with a sleek, modern backdrop displaying **"Spring Collection 2024"**. Models walk confidently on the catwalk, showcasing vibrant, floral prints and pastel tones, under soft, ambient lighting that enhances the fresh, spring vibe; (5) A close-up of a sleek smartwatch on a wrist, the screen displaying **"Step Goal Achieved"** with a celebratory animation, set against a blurred cityscape at dusk, capturing the moment of accomplishment.

## 5 CONCLUSION

In this paper, we showed that DDPO implicitly performs denoising score/flow matching with noisy targets, which increases variance and slows optimization. Building on this, we introduced *Advantage-Weighted Matching* (AWM), which keeps the pretraining score/flow-matching objective and applies advantage-based weights—thereby unifying pretraining and RL post-training under a single, policy-gradient–consistent objective. It achieves up to $24\times$ faster convergence than Flow-GRPO on SD3.5-M and FLUX without degrading generation quality.

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

Table 3: Notations utilized in this paper.

| Notation | Description |
|---|---|
| $\boldsymbol{x}_0$ | Clean data sample from the data distribution |
| $\boldsymbol{x}_t$ | Noisy data at time $t$ |
| $\boldsymbol{c}$ | Conditioning context (e.g., text prompt) |
| $\boldsymbol{\epsilon}$ | Standard Gaussian noise, $\boldsymbol{\epsilon} \sim \mathcal{N}(\mathbf{0}, \boldsymbol{I})$ |
| $t$ | Time step, $t \in [0, 1]$ |
| $\boldsymbol{v}_{\boldsymbol{\theta}}(\boldsymbol{x}_t, t)$ | Velocity prediction network |
| $\boldsymbol{\epsilon}_{\boldsymbol{\theta}}(\boldsymbol{x}_t, t)$ | Noise prediction network |
| $\boldsymbol{s}_{\boldsymbol{\theta}}(\boldsymbol{x}_t, t)$ | Score prediction network |
| $p(\boldsymbol{x})$ | Likelihood of the data |
| $\Delta t$ | Discretization step size for SDE/ODE solvers |
| $\pi_{\boldsymbol{\theta}}$ | Policy model parameterized by $\boldsymbol{\theta}$ |
| $\pi_{\boldsymbol{\theta}_{\text{old}}}$ | Old policy used for sampling (Importance Sampling) |
| $\pi_{\boldsymbol{\theta}_{\text{ref}}}$ | Reference policy for KL regularization (e.g. base model) |
| $r(\boldsymbol{x}, \boldsymbol{c})$ | Reward function evaluating the quality of sample $\boldsymbol{x}$ given $\boldsymbol{c}$ |
| $A$ | Advantage value derived from rewards (e.g., group-relative) |
| $G$ | Group size of samples generated per prompt |
| $\mathcal{J}_{GRPO}$ | Objective function for GRPO |
| $\mathbb{D}_{KL}$ | Kullback-Leibler divergence |
| $\beta$ | Coefficient for the KL regularization term |
| $w(t)$ | Time-dependent weighting function in the loss objective |

## A NOTATION

In this section, we compile a comprehensive list of mathematical notations used throughout the paper to facilitate reference. Given that our work, Advantage Weighted Matching (AWM), bridges the domains of generative diffusion models (specifically Flow Matching) and Reinforcement Learning (RL), we explicitly define symbols for both the generative processes and the policy optimization objectives. A detailed summary of these notations and their corresponding descriptions is provided in Table 3.

## B LLM USAGE

In the preparation of this manuscript, we employed large language models to provide language-related assistance. Specifically, the LLM was used to (i) polish grammar, style, and readability of the text; (ii) offer suggestions for clearer phrasing and more concise expression.

## C PROOFS

### C.1 PROOF OF LEMMA 1

*Proof.* The proof proceeds by taking the gradient of the Denoising Score Matching loss with respect to the model parameters $\boldsymbol{\theta}$ and showing that it is equivalent to the gradient of the standard Score

Matching loss.

$$\nabla_{\boldsymbol{\theta}}\mathbb{E}_{\boldsymbol{x}_s}\mathbb{E}_{\boldsymbol{x}_t|\boldsymbol{x}_s}\left[\|\boldsymbol{s}_{\boldsymbol{\theta}}(\boldsymbol{x}_t, t) - \nabla_{\boldsymbol{x}_t}\log p(\boldsymbol{x}_t|\boldsymbol{x}_s)\|^2\right]$$

$$=\nabla_{\boldsymbol{\theta}}\mathbb{E}_{\boldsymbol{x}_s}\mathbb{E}_{\boldsymbol{x}_t|\boldsymbol{x}_s}\left[\|\boldsymbol{s}_{\boldsymbol{\theta}}(\boldsymbol{x}_t, t)\|^2\right] - 2\nabla_{\boldsymbol{\theta}}\mathbb{E}_{\boldsymbol{x}_s}\mathbb{E}_{\boldsymbol{x}_t|\boldsymbol{x}_s}\left[\langle\boldsymbol{s}_{\boldsymbol{\theta}}(\boldsymbol{x}_t, t), \nabla_{\boldsymbol{x}_t}\log p(\boldsymbol{x}_t|\boldsymbol{x}_s)\rangle\right]$$

$$=\nabla_{\boldsymbol{\theta}}\mathbb{E}_{\boldsymbol{x}_t}\left[\|\boldsymbol{s}_{\boldsymbol{\theta}}(\boldsymbol{x}_t, t)\|^2\right] - 2\nabla_{\boldsymbol{\theta}}\int_{\boldsymbol{x}_t}\int_{\boldsymbol{x}_s}\langle\boldsymbol{s}_{\boldsymbol{\theta}}(\boldsymbol{x}_t, t), \nabla_{\boldsymbol{x}_t}\log p(\boldsymbol{x}_t|\boldsymbol{x}_s)\rangle p(\boldsymbol{x}_t|\boldsymbol{x}_s)p(\boldsymbol{x}_s)\mathrm{d}\boldsymbol{x}_s\mathrm{d}\boldsymbol{x}_t$$

$$=\nabla_{\boldsymbol{\theta}}\mathbb{E}_{\boldsymbol{x}_t}\left[\|\boldsymbol{s}_{\boldsymbol{\theta}}(\boldsymbol{x}_t, t)\|^2\right] - 2\nabla_{\boldsymbol{\theta}}\int_{\boldsymbol{x}_t}\int_{\boldsymbol{x}_s}\langle\boldsymbol{s}_{\boldsymbol{\theta}}(\boldsymbol{x}_t, t), \nabla_{\boldsymbol{x}_t}p(\boldsymbol{x}_t|\boldsymbol{x}_s)\rangle p(\boldsymbol{x}_s)\mathrm{d}\boldsymbol{x}_s\mathrm{d}\boldsymbol{x}_t$$

$$=\nabla_{\boldsymbol{\theta}}\mathbb{E}_{\boldsymbol{x}_t}\left[\|\boldsymbol{s}_{\boldsymbol{\theta}}(\boldsymbol{x}_t, t)\|^2\right] - 2\nabla_{\boldsymbol{\theta}}\int_{\boldsymbol{x}_t}\langle\boldsymbol{s}_{\boldsymbol{\theta}}(\boldsymbol{x}_t, t), \int_{\boldsymbol{x}_s}\nabla_{\boldsymbol{x}_t}p(\boldsymbol{x}_t|\boldsymbol{x}_s)p(\boldsymbol{x}_s)\mathrm{d}\boldsymbol{x}_s\rangle\mathrm{d}\boldsymbol{x}_t$$

$$=\nabla_{\boldsymbol{\theta}}\mathbb{E}_{\boldsymbol{x}_t}\left[\|\boldsymbol{s}_{\boldsymbol{\theta}}(\boldsymbol{x}_t, t)\|^2\right] - 2\nabla_{\boldsymbol{\theta}}\int_{\boldsymbol{x}_t}\langle\boldsymbol{s}_{\boldsymbol{\theta}}(\boldsymbol{x}_t, t), \nabla_{\boldsymbol{x}_t}\int_{\boldsymbol{x}_s}p(\boldsymbol{x}_t|\boldsymbol{x}_s)p(\boldsymbol{x}_s)\mathrm{d}\boldsymbol{x}_s\rangle\mathrm{d}\boldsymbol{x}_t$$

$$=\nabla_{\boldsymbol{\theta}}\mathbb{E}_{\boldsymbol{x}_t}\left[\|\boldsymbol{s}_{\boldsymbol{\theta}}(\boldsymbol{x}_t, t)\|^2\right] - 2\nabla_{\boldsymbol{\theta}}\int_{\boldsymbol{x}_t}\langle\boldsymbol{s}_{\boldsymbol{\theta}}(\boldsymbol{x}_t, t), \nabla_{\boldsymbol{x}_t}p(\boldsymbol{x}_t)\rangle\mathrm{d}\boldsymbol{x}_t$$

$$=\nabla_{\boldsymbol{\theta}}\mathbb{E}_{\boldsymbol{x}_t}\left[\|\boldsymbol{s}_{\boldsymbol{\theta}}(\boldsymbol{x}_t, t)\|^2\right] - 2\nabla_{\boldsymbol{\theta}}\int_{\boldsymbol{x}_t}\langle\boldsymbol{s}_{\boldsymbol{\theta}}(\boldsymbol{x}_t, t), \nabla_{\boldsymbol{x}_t}\log p(\boldsymbol{x}_t)\rangle p(\boldsymbol{x}_t)\mathrm{d}\boldsymbol{x}_t$$

$$=\nabla_{\boldsymbol{\theta}}\mathbb{E}_{\boldsymbol{x}_t}\left[\|\boldsymbol{s}_{\boldsymbol{\theta}}(\boldsymbol{x}_t, t)\|^2\right] - 2\nabla_{\boldsymbol{\theta}}\mathbb{E}_{\boldsymbol{x}_t}\left[\langle\boldsymbol{s}_{\boldsymbol{\theta}}(\boldsymbol{x}_t, t), \nabla_{\boldsymbol{x}_t}\log p(\boldsymbol{x}_t)\rangle\right]$$

$$=\nabla_{\boldsymbol{\theta}}\mathbb{E}_{\boldsymbol{x}_t}\left[\|\boldsymbol{s}_{\boldsymbol{\theta}}(\boldsymbol{x}_t, t) - \nabla_{\boldsymbol{x}_t}\log p(\boldsymbol{x}_t)\|^2\right]$$

$$(13)$$

$\square$

### C.2 PROOF OF THEOREM 1

Our analysis builds upon a foundational theorem regarding the time reversal of diffusion processes from Haussmann and Pardoux (Haussmann & Pardoux, 1986).

**Lemma 2** (Time Reversal of Diffusions (Haussmann & Pardoux, 1986))**.** *Let the forward process* $\{\boldsymbol{x}_t\}_{t\in[0,1]}$ *be defined by the stochastic differential equation (SDE):*

$$\mathrm{d}\boldsymbol{x}_t = -\frac{1}{1-t}\boldsymbol{x}_t, \mathrm{d}t + \sqrt{\frac{2t}{1-t}}, \mathrm{d}\boldsymbol{w}_t, \tag{14}$$

*Then, the corresponding reverse process, where time flows from 1 to 0, satisfies the SDE:*

$$\mathrm{d}\boldsymbol{x}_t = \left[\boldsymbol{v}_{\boldsymbol{\theta}}(\boldsymbol{x}_t, t) + \frac{1}{1-t}\left(\boldsymbol{x}_t + (1-t)\boldsymbol{v}_{\boldsymbol{\theta}}(\boldsymbol{x}_t, t)\right)\right]\mathrm{d}t + \sqrt{\frac{2t}{1-t}}, \mathrm{d}\boldsymbol{w}_t. \tag{15}$$

A direct consequence of this theorem is that the marginal distribution of $\boldsymbol{x}_t$ is identical for the forward and reverse processes. Furthermore, a stronger corollary holds: the joint distribution of $(\boldsymbol{x}_s, \boldsymbol{x}_t)$ is also identical for both processes. This allows us to connect the forward-process-based score matching objective to the reverse-process-based DDPO objective.

*Proof.* Our goal is to demonstrate that minimizing the Denoising Score Matching loss is equivalent to maximizing the DDPO log-likelihood objective. We begin with the Denoising Score Matching loss at $\boldsymbol{x}_t$ conditioned on noisy data $\boldsymbol{x}_{t-\Delta t}$ from the forward process:

$$\mathbb{E}_{\boldsymbol{x}_{t-\Delta t}, \boldsymbol{x}_t}\left[\|\boldsymbol{s}_{\boldsymbol{\theta}}(\boldsymbol{x}_t, t) - \nabla\log p(\boldsymbol{x}_t|\boldsymbol{x}_{t-\Delta t})\|^2\right]. \tag{16}$$

Note that $\boldsymbol{x}_t|\boldsymbol{x}_{t-\Delta t} \sim \mathcal{N}\left(\frac{1-t}{1-(t-\Delta t)}\boldsymbol{x}_{t-\Delta t}, \left(t^2 - \frac{(1-t)^2}{(1-(t-\Delta t))^2}(t - \Delta t)^2\right)\boldsymbol{I}\right)$, thus we have that the above denoising score matching loss is:

$$\mathbb{E}_{\boldsymbol{x}_{t-\Delta t}, \boldsymbol{x}_t}\left[\left\|\boldsymbol{s}_{\boldsymbol{\theta}}(\boldsymbol{x}_t, t) + \frac{\boldsymbol{x}_t - \frac{1-t}{1-(t-\Delta t)}\boldsymbol{x}_{t-\Delta t}}{t^2 - \frac{(1-t)^2}{(1-(t-\Delta t))^2}(t - \Delta t)^2}\right\|^2\right]. \tag{17}$$

Also, note that $s_{\boldsymbol{\theta}}(\boldsymbol{x}_t, t) = -\frac{(1-t)\boldsymbol{v}_{\boldsymbol{\theta}}(\boldsymbol{x}_t, t) + \boldsymbol{x}_t}{t}$, we have

$$\mathbb{E}_{\boldsymbol{x}_{t-\Delta t}, \boldsymbol{x}_t}\left[\left\|-\frac{(1-t)\boldsymbol{v}_{\boldsymbol{\theta}}(\boldsymbol{x}_t, t) + \boldsymbol{x}_t}{t} + \frac{\boldsymbol{x}_t - \frac{1-t}{1-(t-\Delta t)}\boldsymbol{x}_{t-\Delta t}}{t^2 - \frac{(1-t)^2}{(1-(t-\Delta t))^2}(t-\Delta t)^2}\right\|^2\right]. \tag{18}$$

Simplifying the above equation, and let the coefficient of $\boldsymbol{x}_{t-\Delta t}$ to be 1, we have:

$$\mathbb{E}_{\boldsymbol{x}_{t-\Delta t}, \boldsymbol{x}_t}\left[\left\|\boldsymbol{x}_{t-\Delta t} - \left((1-(t-\Delta t)) + \frac{(1-t)(t-\Delta t)^2}{t(1-(t-\Delta t))}\right)\boldsymbol{x}_t\right.\right.$$

$$\left.\left. + \frac{1-(t-\Delta t)}{t}\left(t^2 - \frac{(1-t)^2}{(1-(t-\Delta t))^2}(t-\Delta t)^2\right)\boldsymbol{v}_{\boldsymbol{\theta}}(\boldsymbol{x}_t, t)\right\|^2\right]. \tag{19}$$

Now, let's consider the DDPO objective, which is the log-likelihood of one step of the reverse process. We consider the Euler-Maruyama discretization with Exponential Integrator of the diffusion SDE Equation (1):

$$\mathrm{d}\boldsymbol{x}_t = \left[\boldsymbol{v}_{\boldsymbol{\theta}}(\boldsymbol{x}_t, t) + \frac{1}{1-t}\left(\boldsymbol{x}_t + (1-t)\boldsymbol{v}_{\boldsymbol{\theta}}(\boldsymbol{x}_t, t)\right)\right]\mathrm{d}t + \sqrt{\frac{2t}{1-t}}\mathrm{d}\boldsymbol{w}_t. \tag{20}$$

According to the results in (Xue et al., 2023),

$$\boldsymbol{x}_{t-\Delta t} = \frac{(1-t)(t-\Delta t)^2}{(1-(t-\Delta t))t^2}\boldsymbol{x}_t + \frac{1-(t-\Delta t)}{t^2}(t^2 - \frac{(1-t)^2}{(1-(t-\Delta t))^2}(t-\Delta t)^2)\boldsymbol{x}_{\boldsymbol{\theta}}(\boldsymbol{x}_t, t)$$

$$+ \sqrt{t^2 - \frac{(1-t)^2}{(1-(t-\Delta t))^2}(t-\Delta t)^2}\frac{t-\Delta t}{t}\boldsymbol{\epsilon} \tag{21}$$

Subtitute that $\boldsymbol{x}_{\boldsymbol{\theta}} = \boldsymbol{x}_t - t\boldsymbol{v}_{\boldsymbol{\theta}}$, we have that

$$\boldsymbol{x}_{t-\Delta t} = \left((1-(t-\Delta t)) + \frac{(1-t)(t-\Delta t)^2}{t(1-(t-\Delta t))}\right)\boldsymbol{x}_t \tag{22}$$

$$- \frac{1-(t-\Delta t)}{t}(t^2 - \frac{(1-t)^2}{(1-(t-\Delta t))^2}(t-\Delta t)^2)\boldsymbol{v}_{\boldsymbol{\theta}}(\boldsymbol{x}_t, t)$$

$$+ \sqrt{t^2 - \frac{(1-t)^2}{(1-(t-\Delta t))^2}(t-\Delta t)^2}\frac{t-\Delta t}{t}\boldsymbol{\epsilon} \tag{23}$$

If we omit the discretization error induced by Euler-Maruyama discretization, according to Lemma 2, the joint distribution of $(\boldsymbol{x}_{t-\Delta t}, \boldsymbol{x}_t)$ in the reverse process is identical to $(\boldsymbol{x}_{t-\Delta t}, \boldsymbol{x}_t)$ in the forward process. Then we have that maximizing $\log p_{\boldsymbol{\theta}}(\boldsymbol{x}_{t-\Delta t}|\boldsymbol{x}_t)$ is equivalent to minimizing:

$$\mathbb{E}_{\boldsymbol{x}_{t-\Delta t}, \boldsymbol{x}_t}\left[\left\|\boldsymbol{x}_{t-\Delta t} - \left((1-(t-\Delta t)) + \frac{(1-t)(t-\Delta t)^2}{t(1-(t-\Delta t))}\right)\boldsymbol{x}_t\right.\right.$$

$$\left.\left. + \frac{1-(t-\Delta t)}{t}\left(t^2 - \frac{(1-t)^2}{(1-(t-\Delta t))^2}(t-\Delta t)^2\right)\boldsymbol{v}_{\boldsymbol{\theta}}(\boldsymbol{x}_t, t)\right\|^2\right], \tag{24}$$

which is identical to Equation (19), thus identical to the noisy DSM loss Equation (16). □

## C.3 PROOF OF THEOREM 2

We first give a more detailed version of Theorem 2

**Theorem 2** (Detailed version of Theorem 2 in the main body). *$\boldsymbol{x}_t, \boldsymbol{x}_s$ satisfies*

$$\boldsymbol{x}_t|\boldsymbol{x}_s \sim \mathcal{N}\big(\alpha(t,s)\,\boldsymbol{x}_s,\ \sigma^2(t,s)\,I\big), \qquad \alpha(t,s) = \frac{1-t}{1-s}, \quad \sigma^2(t,s) = t^2 - \frac{(1-t)^2}{(1-s)^2}s^2, \tag{25}$$

*for all $0 \leq s < t \leq 1$, so that*
$$\sigma^2(t,s) + \alpha^2(t,s)\,s^2 \;=\; t^2. \tag{26}$$

*Let $d$ be the data dimension. Then, for the two conditional scores*
$$\nabla_{\boldsymbol{x}_t} \log p(\boldsymbol{x}_t|\boldsymbol{x}_s), \qquad \nabla_{\boldsymbol{x}_t} \log p(\boldsymbol{x}_t|\boldsymbol{x}_0),$$

*and for any fixed $\boldsymbol{x}_t$ and any $s \in [0,t)$, each is an unbiased estimator of the true score* $\nabla_{\boldsymbol{x}_t} \log p(\boldsymbol{x}_t)$:
$$\mathbb{E}[\nabla_{\boldsymbol{x}_t} \log p(\boldsymbol{x}_t|\boldsymbol{x}_s)\,|\,\boldsymbol{x}_t] = \mathbb{E}[\nabla_{\boldsymbol{x}_t} \log p(\boldsymbol{x}_t|\boldsymbol{x}_0)\,|\,\boldsymbol{x}_t] = \nabla_{\boldsymbol{x}_t} \log p(\boldsymbol{x}_t).$$

*Moreover, their conditional covariances satisfy*
$$\mathrm{Cov}(\nabla_{\boldsymbol{x}_t} \log p(\boldsymbol{x}_t|\boldsymbol{x}_s)\,|\,\boldsymbol{x}_t) = \mathrm{Cov}(\nabla_{\boldsymbol{x}_t} \log p(\boldsymbol{x}_t|\boldsymbol{x}_0)\,|\,\boldsymbol{x}_t) + \kappa(s,t)\,I \;\succeq\; \mathrm{Cov}(\nabla_{\boldsymbol{x}_t} \log p(\boldsymbol{x}_t|\boldsymbol{x}_0)\,|\,\boldsymbol{x}_t), \tag{27}$$

*where*
$$\kappa(s,t) \;=\; \frac{(1-t)^2\,s^2}{t^2\left(t^2(1-s)^2 - s^2(1-t)^2\right)}. \tag{28}$$

*In particular,*
$$\mathrm{Tr}\,\mathrm{Cov}(\nabla_{\boldsymbol{x}_t} \log p(\boldsymbol{x}_t|\boldsymbol{x}_s)\,|\,\boldsymbol{x}_t) = \mathrm{Tr}\,\mathrm{Cov}(\nabla_{\boldsymbol{x}_t} \log p(\boldsymbol{x}_t|\boldsymbol{x}_0)\,|\,\boldsymbol{x}_t) + d\,\kappa(s,t).$$

*The map $s \mapsto \kappa(s,t)$ is strictly increasing on $[0,t)$, with $\kappa(0,t) = 0$ and $\kappa(s,t) \to \infty$ as $s \uparrow t$.*

*Moreover, for any predictor $a = a(\boldsymbol{x}_t,t) \in \mathbb{R}^d$ (e.g. $a = \boldsymbol{s}_\theta(\boldsymbol{x}_t,t)$):*
$$\mathbb{E}\Big[\big\|a - \nabla_{\boldsymbol{x}_t} \log p(\boldsymbol{x}_t|\boldsymbol{x}_s)\big\|^2 \,\Big|\, \boldsymbol{x}_t\Big] = \mathbb{E}\Big[\big\|a - \nabla_{\boldsymbol{x}_t} \log p(\boldsymbol{x}_t|\boldsymbol{x}_0)\big\|^2 \,\Big|\, \boldsymbol{x}_t\Big] + d\,\kappa(s,t), \tag{29}$$

$$\mathrm{Var}\Big(\big\|a - \nabla_{\boldsymbol{x}_t} \log p(\boldsymbol{x}_t|\boldsymbol{x}_s)\big\|^2 \,\Big|\, \boldsymbol{x}_t\Big) = \mathrm{Var}\Big(\big\|a - \nabla_{\boldsymbol{x}_t} \log p(\boldsymbol{x}_t|\boldsymbol{x}_0)\big\|^2 \,\Big|\, \boldsymbol{x}_t\Big) + 2d\,\kappa(s,t)^2$$
$$+ 4\,\kappa(s,t)\,\mathbb{E}\Big[\big\|a - \nabla_{\boldsymbol{x}_t} \log p(\boldsymbol{x}_t|\boldsymbol{x}_0)\big\|^2 \,\Big|\, \boldsymbol{x}_t\Big]. \tag{30}$$

*Hence both the conditional risk and the conditional objective variance are minimized at $s = 0$ and strictly increase with $s \in (0,t)$.*

*Proof.* Under Equation (25),
$$\nabla_{\boldsymbol{x}_t} \log p(\boldsymbol{x}_t|\boldsymbol{x}_s) = -\frac{1}{\sigma^2(t,s)}\big(\boldsymbol{x}_t - \alpha(t,s)\,\boldsymbol{x}_s\big), \qquad \nabla_{\boldsymbol{x}_t} \log p(\boldsymbol{x}_t|\boldsymbol{x}_0) = -\frac{1}{t^2}\big(\boldsymbol{x}_t - (1-t)\,\boldsymbol{x}_0\big). \tag{31}$$

**Auxiliary Lemma 1** For fixed $\boldsymbol{x}_t$ and any $s \in [0,t)$, there exists a zero-mean Gaussian vector $\boldsymbol{\eta}$ such that
$$\nabla_{\boldsymbol{x}_t} \log p(\boldsymbol{x}_t|\boldsymbol{x}_s) \;=\; \nabla_{\boldsymbol{x}_t} \log p(\boldsymbol{x}_t|\boldsymbol{x}_0) + \boldsymbol{\eta}, \qquad \boldsymbol{\eta} \perp (\boldsymbol{x}_0|\boldsymbol{x}_t), \qquad \boldsymbol{\eta} \sim \mathcal{N}\big(\boldsymbol{0},\,\kappa(s,t)\,I\big), \tag{32}$$
with $\kappa(s,t)$ given by Equation (28).

*Proof of Aux. Lemma 1.* Conditioned on $(\boldsymbol{x}_t,\boldsymbol{x}_0)$, the posterior of $\boldsymbol{x}_s$ is Gaussian with mean $\frac{\sigma^2(t,s)(1-s)\boldsymbol{x}_0 + \alpha(t,s)s^2\boldsymbol{x}_t}{t^2}$ and covariance $\frac{s^2\sigma^2(t,s)}{t^2}$ Using equation 31, define
$$\boldsymbol{\eta} := \nabla_{\boldsymbol{x}_t} \log p(\boldsymbol{x}_t|\boldsymbol{x}_s) - \nabla_{\boldsymbol{x}_t} \log p(\boldsymbol{x}_t|\boldsymbol{x}_0) = \frac{\alpha(t,s)}{\sigma^2(t,s)}\big(\boldsymbol{x}_s - \mathbb{E}[\boldsymbol{x}_s|\boldsymbol{x}_t,\boldsymbol{x}_0]\big).$$

Then $\boldsymbol{\eta}|(\boldsymbol{x}_t,\boldsymbol{x}_0)$ is zero-mean Gaussian with covariance
$$\frac{\alpha^2}{\sigma^4}\frac{s^2\sigma^2}{t^2}I = \frac{\alpha^2 s^2}{\sigma^2 t^2}\,I = \kappa(s,t)\,I.$$

The residual $\boldsymbol{x}_s - \mathbb{E}[\boldsymbol{x}_s|\boldsymbol{x}_t,\boldsymbol{x}_0]$ is independent of $(\boldsymbol{x}_t,\boldsymbol{x}_0)$, hence $\boldsymbol{\eta} \perp (\boldsymbol{x}_0|\boldsymbol{x}_t)$. $\qquad\square$

By the score identity, $\mathbb{E}[\nabla_{\boldsymbol{x}_t} \log p(\boldsymbol{x}_t|\boldsymbol{x}_s)|\boldsymbol{x}_t] = \nabla_{\boldsymbol{x}_t} \log p(\boldsymbol{x}_t)$ for every $s \in [0,t)$; in particular this equals $\mathbb{E}[\nabla_{\boldsymbol{x}_t} \log p(\boldsymbol{x}_t|\boldsymbol{x}_0)|\boldsymbol{x}_t]$. From equation 32 and $\boldsymbol{\eta} \perp (\boldsymbol{x}_0|\boldsymbol{x}_t)$,
$$\mathrm{Cov}(\nabla_{\boldsymbol{x}_t} \log p(\boldsymbol{x}_t|\boldsymbol{x}_s)\,|\,\boldsymbol{x}_t) = \mathrm{Cov}(\nabla_{\boldsymbol{x}_t} \log p(\boldsymbol{x}_t|\boldsymbol{x}_0)\,|\,\boldsymbol{x}_t) + \mathrm{Cov}(\boldsymbol{\eta}|\boldsymbol{x}_t)$$
$$= \mathrm{Cov}(\nabla_{\boldsymbol{x}_t} \log p(\boldsymbol{x}_t|\boldsymbol{x}_0)\,|\,\boldsymbol{x}_t) + \kappa(s,t)\,I,$$

which proves Theorem 2 and the trace statement.

**Auxiliary Lemma 2 (Monotonicity of $\kappa$).** For fixed $t \in (0,1]$, $s \mapsto \kappa(s,t)$ in Equation (28) is strictly increasing on $[0,t)$, with $\kappa(0,t) = 0$ and $\kappa(s,t) \to \infty$ as $s \uparrow t$.

*Proof of Aux. Lemma 2.* Using Equation (26), write $\kappa(s,t) = \frac{q(s)}{t^2\left(t^2 - q(s)\right)}$ with $q(s) := \alpha^2(t,s)s^2 = \frac{(1-t)^2 s^2}{(1-s)^2}$. On $[0,t)$, $q(s)$ is strictly increasing and takes values in $[0,t^2)$, while $q \mapsto \frac{q}{t^2(t^2-q)}$ is strictly increasing on $[0,t^2)$. Limits follow from $q(0) = 0$ and $q(s) \uparrow t^2$ as $s \uparrow t$. $\qquad\square$

For any $a$,

$$\mathbb{E}\Big[\big\|a - \nabla_{\boldsymbol{x}_t}\log p(\boldsymbol{x}_t|\boldsymbol{x}_s)\big\|^2 \,\Big|\, \boldsymbol{x}_t\Big] = \Big\|a - \mathbb{E}[\nabla_{\boldsymbol{x}_t}\log p(\boldsymbol{x}_t|\boldsymbol{x}_s)|\boldsymbol{x}_t]\Big\|^2 + \operatorname{Tr}\operatorname{Cov}(\nabla_{\boldsymbol{x}_t}\log p(\boldsymbol{x}_t|\boldsymbol{x}_s) \mid \boldsymbol{x}_t).$$

Insert unbiasedness and Equation (27), then subtract the same identity with $s = 0$ to obtain the increment $d\,\kappa(s,t)$.

To prove Equation (30), let $\boldsymbol{\delta}(\boldsymbol{x}_0) := a - \nabla_{\boldsymbol{x}_t}\log p(\boldsymbol{x}_t|\boldsymbol{x}_0)$. From Equation (32),

$$\big\|a - \nabla_{\boldsymbol{x}_t}\log p(\boldsymbol{x}_t|\boldsymbol{x}_s)\big\|^2 = \|\boldsymbol{\delta} - \boldsymbol{\eta}\|^2 = \|\boldsymbol{\delta}\|^2 + \|\boldsymbol{\eta}\|^2 - 2\langle\boldsymbol{\delta},\boldsymbol{\eta}\rangle.$$

Conditioned on $(\boldsymbol{x}_t, \boldsymbol{x}_0)$, $\boldsymbol{\eta} \sim \mathcal{N}(\mathbf{0}, \kappa I)$ is independent of $\boldsymbol{\delta}$. Gaussian moment identities yield, coordinatewise,

$$\mathbb{E}[\|\boldsymbol{\eta}\|^2|\boldsymbol{x}_t,\boldsymbol{x}_0] = d\,\kappa, \quad \operatorname{Var}(\|\boldsymbol{\eta}\|^2|\boldsymbol{x}_t,\boldsymbol{x}_0) = 2d\,\kappa^2,$$

$$\operatorname{Var}(\langle\boldsymbol{\delta},\boldsymbol{\eta}\rangle|\boldsymbol{x}_t,\boldsymbol{x}_0) = \kappa\,\|\boldsymbol{\delta}\|^2, \quad \operatorname{Cov}(\|\boldsymbol{\eta}\|^2, \langle\boldsymbol{\delta},\boldsymbol{\eta}\rangle|\boldsymbol{x}_t,\boldsymbol{x}_0) = 0.$$

Therefore,

$$\operatorname{Var}\Big(\big\|a - \nabla_{\boldsymbol{x}_t}\log p(\boldsymbol{x}_t|\boldsymbol{x}_s)\big\|^2 \,\Big|\, \boldsymbol{x}_t, \boldsymbol{x}_0\Big) = \operatorname{Var}(\|\boldsymbol{\eta}\|^2 - 2\langle\boldsymbol{\delta},\boldsymbol{\eta}\rangle|\boldsymbol{x}_t,\boldsymbol{x}_0) = 2d\,\kappa^2 + 4\kappa\,\|\boldsymbol{\delta}\|^2.$$

Law of total variance yields Equation (30). Monotonicity in $s$ follows from Aux. Lemma 2; strictness for $s \in (0,t)$ follows since $\kappa(s,t) > 0$. $\qquad\square$

### C.4 DEVIATION OF EQUATION (8)

In this section, we provide the detailed derivation of the noisy Denoising Score Matching (DSM) objective under the EDM parameterization (Equation 8).

**1. Conditional Distribution** $p(\boldsymbol{x}_t|\boldsymbol{x}_s)$**.** The EDM noise schedule is defined as $\boldsymbol{x}_t = \boldsymbol{x}_0 + t\boldsymbol{\epsilon}$, where $\boldsymbol{\epsilon} \sim \mathcal{N}(\mathbf{0}, \boldsymbol{I})$. For $0 \le s < t$, the transition from $\boldsymbol{x}_s$ to $\boldsymbol{x}_t$ follows a Gaussian distribution due to the properties of the equivalent variance-exploding SDE. The conditional distribution is given by:

$$p(\boldsymbol{x}_t|\boldsymbol{x}_s) = \mathcal{N}(\boldsymbol{x}_t; \boldsymbol{x}_s, (t^2 - s^2)\boldsymbol{I}). \tag{33}$$

**2. Conditional Score** $\nabla_{\boldsymbol{x}_t}\log p(\boldsymbol{x}_t|\boldsymbol{x}_s)$**.** The score function corresponding to this conditional distribution is the gradient of the log-likelihood with respect to $\boldsymbol{x}_t$:

$$\nabla_{\boldsymbol{x}_t}\log p(\boldsymbol{x}_t|\boldsymbol{x}_s) = \nabla_{\boldsymbol{x}_t}\left(-\frac{1}{2(t^2 - s^2)}\|\boldsymbol{x}_t - \boldsymbol{x}_s\|^2 + \text{const}\right)$$

$$= -\frac{\boldsymbol{x}_t - \boldsymbol{x}_s}{t^2 - s^2}. \tag{34}$$

**3. Relation between Denoiser $\boldsymbol{D}_{\boldsymbol{\theta}}$ and Score.** In the EDM framework, the model is parameterized as a denoiser $\boldsymbol{D}_{\boldsymbol{\theta}}(\boldsymbol{x}_t, t)$ trained to predict the clean data $\boldsymbol{x}_0$. The relationship between the optimal denoiser and the score function is (Tweedie's formula):

$$\boldsymbol{D}_{\boldsymbol{\theta}}(\boldsymbol{x}_t, t) = \boldsymbol{x}_t + t^2\nabla_{\boldsymbol{x}_t}\log p(\boldsymbol{x}_t). \tag{35}$$

In the context of Noisy DSM, we substitute the true score $\nabla_{\boldsymbol{x}_t}\log p(\boldsymbol{x}_t)$ with the conditional score $\nabla_{\boldsymbol{x}_t}\log p(\boldsymbol{x}_t|\boldsymbol{x}_s)$ as the regression target.

**4. Target for $D_{\boldsymbol{\theta}}$.** By substituting the conditional score into the denoiser equation, we obtain the training target for $D_{\boldsymbol{\theta}}(\boldsymbol{x}_t, t)$:

$$
\begin{aligned}
\text{Target}(\boldsymbol{x}_t, \boldsymbol{x}_s) &= \boldsymbol{x}_t + t^2 \left( \nabla_{\boldsymbol{x}_t} \log p(\boldsymbol{x}_t | \boldsymbol{x}_s) \right) \\
&= \boldsymbol{x}_t + t^2 \left( -\frac{\boldsymbol{x}_t - \boldsymbol{x}_s}{t^2 - s^2} \right) \\
&= \boldsymbol{x}_t - \frac{t^2}{t^2 - s^2} (\boldsymbol{x}_t - \boldsymbol{x}_s).
\end{aligned}
\tag{36}
$$

Therefore, the Noisy DSM loss function in terms of the denoiser $D_{\boldsymbol{\theta}}$ is:

$$
\mathcal{L}_{\text{Noisy-DSM}} = \mathbb{E}_{\boldsymbol{x}_s, \boldsymbol{x}_t} \left[ w(t) \left\| D_{\boldsymbol{\theta}}(\boldsymbol{x}_t, t) - \left( \boldsymbol{x}_t - \frac{t^2}{t^2 - s^2} (\boldsymbol{x}_t - \boldsymbol{x}_s) \right) \right\|^2 \right],
\tag{37}
$$

which matches Equation 8.

# D    DETAILED DESCRIPTION OF DDPO

Denoising diffusion policy optimization (DDPO) (Black et al., 2023) map the iterative denoising procedure to the following MDP:

**Problem Setup of Denoising Diffusion Policy Optimization**

$$
\text{State:} \quad \boldsymbol{s}_t = (\boldsymbol{c}, t, \boldsymbol{x}_t) \qquad \text{Action:} \quad \boldsymbol{a}_t = \boldsymbol{x}_{t-1} \qquad \text{Policy:} \quad \pi(\boldsymbol{a}_t | \boldsymbol{s}_t) = p_{\boldsymbol{\theta}}(\boldsymbol{x}_{t-1} | \boldsymbol{x}_t, \boldsymbol{c})
$$

To calculate the reverse transition probability $p_{\boldsymbol{\theta}}(\boldsymbol{x}_{t-1} | \boldsymbol{x}_t, \boldsymbol{c})$, we first consider the sampling process of diffusion or flow matching models involves solving the reverse time diffusion SDE (Song et al., 2020), where $\boldsymbol{w}_t$ represents the Wiener process:

$$
\mathrm{d}\boldsymbol{x}_t = \left[ \boldsymbol{v}_{\boldsymbol{\theta}}(\boldsymbol{x}_t, t) + \frac{1}{1-t} \left( \boldsymbol{x}_t + (1-t)\boldsymbol{v}_{\boldsymbol{\theta}}(\boldsymbol{x}_t, t) \right) \right] \mathrm{d}t + \sqrt{\frac{2t}{1-t}} \mathrm{d}\boldsymbol{w}_t,
$$

However, the exact transition probability $p_{\boldsymbol{\theta}}(\boldsymbol{x}_{t-1} | \boldsymbol{x}_t, \boldsymbol{c})$ is analytically intractable. To address this, DDPO implements the Euler–Maruyama discretization of the reverse SDE:

$$
\boldsymbol{x}_{t-\Delta t} = \boldsymbol{x}_t - \left[ \boldsymbol{v}_{\boldsymbol{\theta}}(\boldsymbol{x}_t, t) + \frac{1}{1-t} \left( \boldsymbol{x}_t + (1-t)\boldsymbol{v}_{\boldsymbol{\theta}}(\boldsymbol{x}_t, t) \right) \right] \Delta t + \sqrt{\frac{2t}{1-t}} \sqrt{\Delta t} \boldsymbol{\epsilon},
$$

Then under the discretization, the transition probability $p_{\boldsymbol{\theta}}(\boldsymbol{x}_{t-1} | \boldsymbol{x}_t, \boldsymbol{c}) \sim \mathcal{N}(, \sigma^2 I)$ becomes an isotropoc Gaussian with tractable mean $\boldsymbol{\mu}_{\boldsymbol{\theta}}$ and variance $\sigma^2$:

$$
\boldsymbol{\mu}_{\boldsymbol{\theta}} = \boldsymbol{x}_t - \left[ \boldsymbol{v}_{\boldsymbol{\theta}}(\boldsymbol{x}_t, t) + \frac{1}{1-t} \left( \boldsymbol{x}_t + (1-t)\boldsymbol{v}_{\boldsymbol{\theta}}(\boldsymbol{x}_t, t) \right) \right] \Delta t, \quad \sigma^2 = \frac{2t}{1-t} \Delta t,
$$

yielding the per-step log-likelihood $\log p_{\boldsymbol{\theta}}(\boldsymbol{x}_{t-\Delta t} | \boldsymbol{x}_t)$:

$$
- \left\| \boldsymbol{x}_{t-\Delta t} - \left( \boldsymbol{x}_t - \left[ \boldsymbol{v}_{\boldsymbol{\theta}}(\boldsymbol{x}_t, t) + \frac{1}{1-t} \left( \boldsymbol{x}_t + (1-t)\boldsymbol{v}_{\boldsymbol{\theta}}(\boldsymbol{x}_t, t) \right) \right] \Delta t \right) \right\|^2 / \left( \frac{4t}{1-t} \Delta t \right) + \text{const.}
$$

Then the DDPO's objective is

$$
\mathbb{E}_{p_{\boldsymbol{\theta}_{\text{old}}}} \left[ \sum_{t=0}^{T} \frac{p_{\boldsymbol{\theta}}(\boldsymbol{x}_{t-1} | \boldsymbol{x}_t, \boldsymbol{c})}{p_{\boldsymbol{\theta}_{\text{old}}}(\boldsymbol{x}_{t-1} | \boldsymbol{x}_t, \boldsymbol{c})} r(\boldsymbol{x}_0, \boldsymbol{c}) \right].
$$

Under the GRPO (Shao et al., 2024) framework, the reward $r(\boldsymbol{x}_0, \boldsymbol{c})$ is substitued to advantage $A(\boldsymbol{x}_0, \boldsymbol{c})$, where the advantage is calculated via $A(\boldsymbol{x}_0, \boldsymbol{c}) = \frac{r(\boldsymbol{x}_0, \boldsymbol{c}) - \text{mean}_{i=0, \cdots, G-1} \{ r(\boldsymbol{x}_i, \boldsymbol{c}) \}}{\text{std}_{i=0, \cdots, G-1} \{ r(\boldsymbol{x}_i, \boldsymbol{c}) \}}$.

## E RELATED WORK

### E.1 REWARD FEEDBACK LEARNING

Imagereward (Xu et al., 2023) proposes Reward Feedback Learning (ReFL), which directly maximizes the reward of an approximately one-step predicted image via gradient backpropagation. To mitigate the approximation error of the one-step prediction, DRaFT (Clark et al., 2023) maximizes the reward of the final multi-step sampled images. While this incurs a massive memory cost, DRaFT reduces it through backpropagation truncation and gradient checkpointing. Viewing the problem from a continuous-time perspective, Adjoint Matching (Domingo-Enrich et al., 2024) applies the adjoint method (Pontryagin, 2018) for memory-efficient gradient computation. There are also trials of applying Reward Feedback Learning on few-step distilled models (Kim et al., 2024; Li et al., 2024; Luo et al., 2024; 2025).

However, the Reward Feedback Learning method has some inevitable drawbacks. First, the reward function must be differentiable, which prevents the use of rule-based or binary rewards, such as those from detection models like GenEval (Ghosh et al., 2023) or OCR. Second, modern generative models operate in a highly compressed VAE latent space. Backpropagating gradients from pixel-level rewards through the VAE decoder adds non-negligible memory and computation costs. Third, although first-order optimization is generally more efficient than zero-order optimization (Nocedal & Wright, 2006), it carries a higher risk of reward hacking, where the model exploits the reward function in unintended ways.

### E.2 DENOISING DIFFUSION POLICY OPTIMIZATION

DDPO (Black et al., 2023) and DPOK (Fan et al., 2023) frame the reinforcement learning problem as a multi-step decision-making problem. They consider an Euler-Maruyama discretization of the reverse process Equation (1), leading to a tractable Gaussian likelihood at each step. DeepSeek-Math (Shao et al., 2024) proposed Group Relative Policy Optimization (GRPO) by substituting the baseline from an additional Value model in PPO (Schulman et al., 2017) to a group relative mean of reward. Flow-GRPO (Liu et al., 2025) and DanceGRPO (Xue et al., 2025) combine the DDPO formulation and use GRPO as its RL algorithm. TempFlow (He et al., 2025) introduces a trajectory branching mechanism that provides process rewards by concentrating stochasticity at designated branching points. MixGRPO (Li et al., 2025a) leverages the flexibility of mixed sampling strategies through the integration of ODE and SDE sampling. BranchGRPO (Li et al., 2025b) introduces a branching scheme that amortizes rollout cost through shared prefixes while preserving exploration diversity.

### E.3 REWARD WEIGHTED REGRESSION

Lee et al. (2023) proposes to finetune a text-to-image model by maximizing an offline reward-weighted denoising loss. Fan et al. (2025) considers an online version of reward-weighted denoising loss with a Wasserstein-2 regularization. Concurrent to ours, FMPG (McAllister et al., 2025) proposes using the ELBO as a proxy for policy likelihood. Unlike our approach, they do not articulate the connection to DDPO, and they do not present experiments on text-to-image diffusion.

### E.4 CONTINUOUS-TIME RL AND CONTROL FOR DIFFUSION

Gao et al. (2024) cast reward-directed diffusion as an entropy-regularized continuous-time RL problem with a running penalty that pushes the denoising dynamics toward the (unknown) true score. They avoid relying on a pretrained score by using a simple ratio estimator to obtain noisy score observations, and solve the problem with a Gaussian-policy q-learning algorithm.

Zhao et al. (2025) treat the score itself as the *action* in a continuous-time stochastic control/RL objective, optimizing terminal reward with a time-integrated KL regularizer between the fine-tuned and pretrained distributions; they derive continuous-time policy-gradient results and report robustness to time discretization.

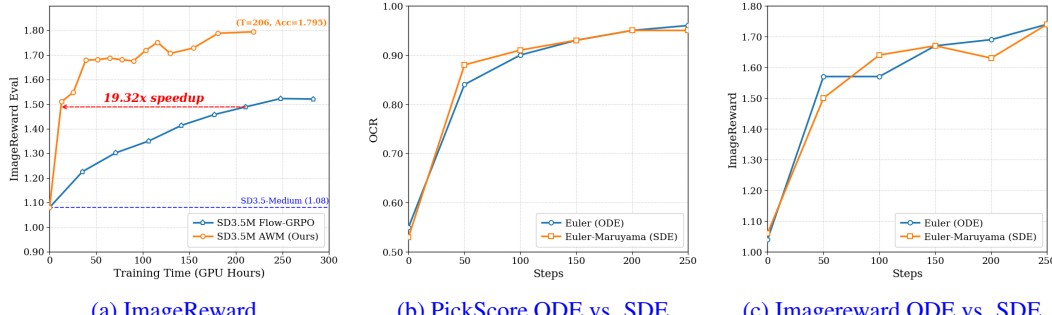

(a) ImageReward  (b) PickScore ODE vs. SDE  (c) Imagereward ODE vs. SDE

Figure 8: (a) Evaluation ImageReward on SD3.5-Medium, comparing AWM (ours) with Flow-GRPO. AWM achieves stronger performance with substantially lower compute. (b) Training-time PickScore rewards for AWM on SD3.5-Medium under Euler and Euler–Maruyama samplers. (c) Training-time ImageReward curves for AWM under the same two samplers. Together, (b)–(c) illustrate AWM's sampler-agnostic behavior: despite differing numerical solvers, AWM yields consistent and stable reward improvement.

## F  ADDITIONAL EXPERIMENTS

In addition to the main results, we conduct several supplementary experiments on SD3.5-Medium to further validate the efficiency and robustness of AWM. First, we compare AWM with Flow-GRPO under the ImageReward metric and observe that AWM achieves substantially higher evaluation reward with significantly less compute, demonstrating its superior sample efficiency, as shown in Figure 8(a).

We also investigate the impact of different numerical samplers during training. Specifically, we evaluate AWM under both the deterministic Euler ODE solver and the stochastic Euler–Maruyama SDE solver for two reward types—PickScore and ImageReward. Figure 8(b)(c) shows that, across both metrics, the training curves exhibit highly consistent improvements, indicating that AWM is sampler-agnostic: it retains stable optimization behavior even when the underlying diffusion sampler changes. This insensitivity to solver choice highlights the practical robustness and broad applicability of AWM in diffusion model RL finetuning.

## G  ADDITIONAL SAMPLES

As shown in Figure 9, even after 600 RL optimization steps, our method preserves broad mode coverage: four samples generated from identical prompts but different seeds remain visually diverse rather than collapsing into a single mode.

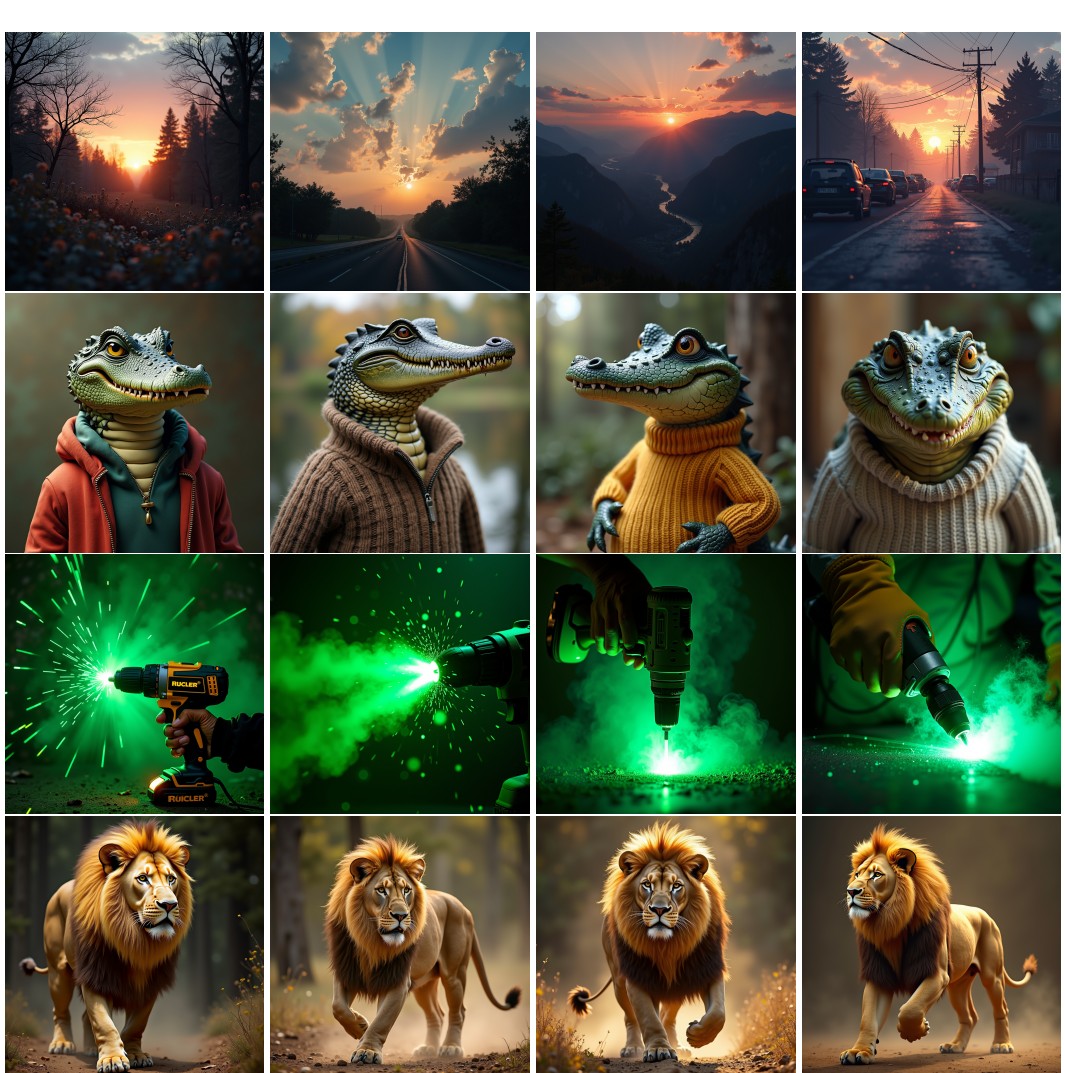

Figure 9: FLUX after 600 RL steps. Using the same prompt, four different seeds (initial noise) are sampled for each row. Our method maintains strong sample diversity. **Row 1**: "Amid the soft glow of twilight." **Row 2**: "Crocodile in a sweater." **Row 3**: "A vivid nuclear green electric drill in action." **Row 4**: "A regal lion."

