# OpenReview forum: "Advantage Weighted Matching: Aligning RL with Pretraining in Diffusion Models"
_ICLR.cc/2026/Conference — Submitted to ICLR 2026_

### Official Review · Reviewer_FSZB · 2025-10-27

**Soundness:** 3
**Presentation:** 4
**Contribution:** 2
**Rating:** 4
**Confidence:** 4

**Summary:**

This paper proposes Advantage Weighted Matching (AWM), a method for applying reinforcement learning to diffusion models by directly optimizing the pre-training score/flow matching objective. The central claim is that prior methods like Denoising Diffusion Policy Optimization (DDPO) are inefficient because they implicitly optimize a score-matching objective with noisy targets, which leads to high-variance gradients. The authors' proposed solution, AWM, avoids this issue by re-weighting the standard "clean-data" score matching loss with an advantage term derived from a policy-gradient algorithm. The paper demonstrates that this approach leads to significant wall-clock speedups (up to 24x) over a Flow-GRPO baseline on benchmarks for compositionality, text rendering, and human preference, while achieving similar final performance.

**Strengths:**

The paper is well-written, and the proposed method is empirically effective. The primary strengths are:

Strong Empirical Results:  the paper demonstrates significant training speed-ups. An 8-24x reduction in GPU hours is a substantial engineering improvement that makes RL fine-tuning for diffusion models more practical and accessible. The experiments are conducted on relevant, modern architectures (SD3.5M, FLUX) and standard benchmarks.

The paper offers a plausible explanation for the inefficiency of DDPO by linking it to a high-variance, noisy-target score-matching objective.

**Weaknesses:**

Despite the impressive speed-up numbers, there are significant concerns regarding the paper's overall contribution and the soundness of its core claims:

Limited Novelty of the Core Idea: The central mechanism of AWM—multiplying a regression loss by a reward-based weight—is a direct application of the well-known Reward-Weighted Regression (RWR) principle. The paper acknowledges prior and concurrent work (e.g., Lee et al. 2023, McAllister et al. 2025) that proposes nearly identical mechanisms. The contribution of this paper appears to be an application of this known technique to an online policy-gradient framework, which feels more incremental than foundational. The novelty seems to lie more in the theoretical "framing" of the problem rather than in the method itself.

Significant Discrepancies Between Theory and Practice: The theoretical argument, while elegant, shows some critical disconnects from the practical implementation, undermining its explanatory power:
The authors state that AWM "unifies pretraining and RL," yet they find that the theoretically-grounded ELBO weighting from pre-training must be discarded in favor of a simple uniform weighting (w(t)=1) to achieve good results. This suggests the theoretical alignment is weaker than claimed and that the method's success relies on an empirical heuristic rather than the core theory.
The main theoretical result (Theorem 1) relies on "omitting the discretization error." This is a strong assumption that sidesteps the complexities of the discrete-time sampling process that all practical models use. This simplification weakens the claim that the theory accurately describes the behavior of real-world implementations.

Insufficient Experimental Depth: The experiments focus heavily on training speed but lack depth in other critical areas. There is no qualitative analysis of failure modes, which is essential for understanding the trade-offs of the new method. For example, does the more direct optimization of AWM lead to a reduction in output diversity or make it more susceptible to reward hacking? Without this analysis, it is difficult to assess whether the speed-up comes at a hidden cost. The comparison to Flow-GRPO could also be strengthened by providing more details on the hyperparameter tuning for the baseline.

**Questions:**

Given the conceptual similarity to Reward-Weighted Regression, could the authors please clarify what they see as the primary novel contribution of AWM beyond applying this principle to an online RL setup for diffusion models? Why should this be considered more than an effective application of a known technique?

The disconnect between the ELBO theory and the practical success of uniform time-weighting is a major concern. Does this not undermine the central claim that AWM successfully "realigns" RL with the pre-training objective? Could the success of AWM be less about aligning with the pre-training likelihood objective and more about the general stability of directly optimizing a single-step objective compared to DDPO's multi-step approach?

---

> ### Author Response · Authors · 2025-11-24
>
> **Weaknesses (1):** AWM’s core mechanism parallels established Reward-Weighted Regression (RWR), with novelty mainly in framing rather than algorithmic substance.
>
> **Response (1):** We respectfully disagree that AWM is merely a direct application of standard Reward-Weighted Regression (RWR). While the naming is similar, AWM is conceptually and technically different from the RWR variants in Lee et al. [1] and Fan et al. [2].
>
> **1. Difference in Weighting Mechanism**
> Existing RWR methods rely on an **Importance Sampling** or **Density Matching** perspective, which necessitates **strictly positive weights** to define a valid probability distribution.
> *   **Lee et al. [1]** use raw reward $r \in [0,1]$ as weights to minimize MSE. This is validated in their paper's section 3.2, "It is trained to predict human feedback $y \in \{0,1\}$", "$r_\phi$ is trained by minimizing the MSE loss $(y - r_\phi)^2$".
> *   **Fan et al. [2]** derive their methods from a **importance sampling** perspective, using weights proportional to strictly positive weight $r$, as validated in their **Theorem 1 and Theorem 2** in [2]. They consider the normalizing constant $Z = \int w(x)q(x) dx$, where $w(x)\propto r(x)$. This requires their $r(x)$ to be **strictly positive**.
> * In [2] and [3], the authors instantiate reward-weighted regression using weights of the form $\exp(T r)$, with temperature $T$ and reward $r$. For advantage-weighted regression, they analogously adopt weights $\exp(T A)$, where $A$ is the advantage. Consequently, the “reward/advantage weighting” used in these papers is explicitly **exponential weighting**, guaranteeing that all assigned weights remain **strictly positive**.
> *  **FMPG (Flow-Matching Policy Gradient)** is concurrent to our work and algorithmically related. However, FMPG does **not analyze the connection to DDPO**, nor does it evaluate on text-to-image diffusion tasks. Instead, its focus remains on conventional RL environments.
>
> In contrast, AWM directly use the **linear advantage $A$** as the weight, which can be **negative**, and is derived from a policy gradient perspective rather than density matching / importance sampling.
>
> Exponential/positive weighting (previous works) only "pulls" the model toward good samples. AWM pulls the model toward samples with $A>0$ and **actively pushes** the model away from samples with $A<0$. This linear weighting is consistent with the standard Policy Gradient objective $\mathbb{E}[\nabla \log \pi \cdot A]$, making AWM mathematically distinct from the density-matching RWR approaches.
>
> **2. Theoretical Contribution: Diagnosing and Fixing DDPO**
> Our contribution is not just "framing", but using theory to diagnose a fundamental inefficiency in existing RL methods. Specifically:
> *   We prove that **DDPO is implicitly performing denoising score matching with noisy targets**, which introduces high variance.
> *   This theoretical insight directly dictates the method: AWM is the "clean-target" correction of DDPO. This insight is absent in prior works, which treat DDPO and regression as separate paradigms.
>
> **3. First Integration with GRPO**
> AWM is the first to integrate **Group Relative Policy Optimization (GRPO)** into regression-based training for diffusion models, which is not covered in [1–3].
>
> [1] Aligning Text-to-Image Models using Human Feedback. Kimin Lee, etc. 2023.
>
> [2] Online Reward-Weighted Fine-Tuning of Flow Matching with Wasserstein Regularization. Jiajun Fan, etc. 2025.
>
> [3] Fine-tuning Flow Matching Generative Models with Intermediate Feedback. Jiajun Fan, etc. 2025.

---

> > ### Author Response · Authors · 2025-11-24
> >
> > **Weaknesses (2):** Theory–practice inconsistency: claimed “unification” with pretraining breaks down as ELBO weights must be replaced by uniform weights; main theorem depends on omitting discretization error, weakening its applicability to real diffusion models.
> >
> > **Response (2):**  **As for the weighting,** we appreciate the reviewer raising this important point regarding the consistency. We would like to clarify this design choice from two perspectives:
> >
> > - *1. Alignment with Established Pretraining Practices*
> > As established in foundational works [4, 5, 6], while the ELBO weighting theoretically optimizes log-likelihood, it **allocates excessive capacity to high-frequency** (small $t$), leading to **suboptimal visual quality and FID scores**. Consequently, more uniform weighting schemes has become the **de facto standard for pretraining** to balance visual quality across all noise levels.
> >
> > - *2. Invariance of the Optimal Solution*
> > Theoretically, the weighting function $w(t)$ serves as a task-balancing mechanism. As noted in [7], if the model has sufficient capacity (or if parameters are not shared across t in the model), the **global optimum** of the score matching objective is invariant to the choice of $w(t)$. Therefore, changing the weight alters the optimization trajectory and convergence speed, especially in the current weight sharing case, but does not theoretically change the optimal objective.
> >
> >
> >
> > **As for the "*discretization error*",** the reviewer correctly notes that the equivalence in Theorem 1 holds up to the discretization error. We emphasize that this is not a simplification in our analysis, but rather underscores a **core, inherent flaw of the DDPO algorithm itself.**
> >
> > * **DDPO's objective relies on discretization:** The DDPO objective is, by definition, the log-likelihood of a *specific, discrete, reverse-time transition* $\log p(x_{t-1}|x_t)$. This means DDPO's training objective is tied to the discretization, as $x_{t-1}|x_t$ approximates Gaussian only when the time gap is small[8].
> > * **Role of the forward process in AWM:** In contrast, the forward transition $x_t \mid x_{t-1}$ is defined analytically and therefore does not incur numerical discretization error. In Theorem 1, the only approximation arises when $(x_{t-1}, x_t)$ is simulated via the Euler–Maruyama discretization of the reverse process; drawing $(x_{t-1}, x_t)$ from the forward process remains exact. This is the sole source of discrepancy in Theorem 1.
> > * **Conclusion:** Thus, the "*discretization error*" is **not an analytical oversight** but **an inherent additional error present in DDPO** and avoided by AWM. The faster convergence of AWM is, in part, due to it avoids the discretization error that burdens DDPO.
> >
> > **References:**
> >
> > [4] Denoising Diffusion Probabilistic Models, Jonathan Ho, etc. 2020.
> >
> > [5] Improved Denoising Diffusion Probabilistic Models, Alex Nichol, etc. 2021.
> >
> > [6] Maximum Likelihood Training of Score-Based Diffusion Models, Yang Song, etc. 2021.
> >
> > [7] Denoising Diffusion Implicit Models, Jiaming Song, etc. 2020.
> >
> > [8] Tackling the Generative Learning Trilemma with Denoising Diffusion GANs. Zhisheng Xiao, etc. 2022.

---

> ### Author Response · Authors · 2025-11-24
>
> **Weaknesses (3):** Experiments focus on speed but omit crucial analyses such as failure modes, diversity trade-offs, and susceptibility to reward hacking; comparisons to Flow-GRPO lack hyperparameter-tuning transparency.
>
> **Response (3):**  We appreciate the reviewer’s suggestion to deepen the experimental analysis.
>
> **1. Fair Comparison: Hyperparameter Settings**
> To address the concern regarding baseline tuning, we emphasize that we aligned all critical hyperparameters to ensure a strictly fair comparison. We did not favor AWM; in fact, we used the standard optimized settings reported in the Flow-GRPO paper (Liu et al., 2025) and applied them identically to AWM where applicable.
>
> | Hyperparameter | Flow-GRPO (Baseline) | AWM (Ours) | Note |
> | :--- | :--- | :--- | :--- |
> | **Learning Rate** | $3 \times 10^{-4}$ | $3 \times 10^{-4}$ | Identical settings. |
> | **LoRA Rank** | $r=64$ (SD3.5M) / $128$ (FLUX) | $r=64$ (SD3.5M) / $128$ (FLUX) | Identical capacity. |
> | **Sampler** | Euler-Maruyama | Euler-Maruyama | Matched for fairness. |
> | **Group Size ($G$)** | 24 | 24 | Identical exploration. |
> | **Reward Function** | GenEval / OCR / PickScore | GenEval / OCR / PickScore | Identical reward function. |
>
>
> **2. Analysis of Failure Modes (Mode Collapse)**
> We explicitly analyzed failure modes in **Section 5.2 (Ablation Studies)** and **Figure 7b**.
> *   **Mode Collapse:** We identified that AWM, like other RL methods, is susceptible to mode collapse if the KL regularization is too weak.
> *   **Mitigation:** We show that a moderate KL penalty ($\beta \in [0.4, 1.0]$) effectively mitigates this risk, balancing reward maximization with the original data distribution constraints.
>
>
> **3. Analysis on Diversity**
> We additionally checked whether AWM sacrifices diversity by running multiple seeds on a random subset of prompts using the AWM-finetuned FLUX model. The generated images show clear variation in both content and style across seeds, confirming that our method does not compromise generation diversity. The results have been updated in Appendix G.
>
> ---
>
> **Questions (1):** Given the conceptual similarity to Reward-Weighted Regression, could the authors please clarify what they see as the primary novel contribution of AWM beyond applying this principle to an online RL setup for diffusion models? Why should this be considered more than an effective application of a known technique?
>
> **Response (1):** See responses to **Weakness (1)**.
>
> ---
>
> **Questions (2):** The disconnect between the ELBO theory and the practical success of uniform time-weighting is a major concern. Does this not undermine the central claim that AWM successfully "realigns" RL with the pre-training objective? Could the success of AWM be less about aligning with the pre-training likelihood objective and more about the general stability of directly optimizing a single-step objective compared to DDPO's multi-step approach?
>
> **Response (1):** See responses to **Weakness (2)**.

---

### Official Review · Reviewer_ZC3a · 2025-10-29

**Soundness:** 2
**Presentation:** 2
**Contribution:** 2
**Rating:** 2
**Confidence:** 4

**Summary:**

This paper introduces Advantage Weighted Matching (AWM), a training algorithm for the reinforcement learning (RL) fine-tuning of diffusion models.  The authors note that Denoising Diffusion Policy Optimization (DDPO) implicitly performs score matching with noisy targets, which increases variance and slows down convergence. AWM is then proposed to address this issue by applying advantage-weighted score/flow matching. Experimental results show that AWM achieves significant speedups over Flow-GRPO.

**Strengths:**

1. The paper discusses the relationship and connection between DDPO  and DSM (Denoising Score Matching).

2. Experimental experiments show that the proposed AWM method achieves acceleration across multiple models, such as SD3.5M and FLUX.

3. It is acknowledged in Appendix C.1.3 that AWM uses the same ELBO policy proxy  as  the work FPO (FMPG), referred to as concurrent work, but differs in its theoretical motivation and experiments.

**Weaknesses:**

1.  The paper repeatedly claims that AWM  *decouples training from sampling* and *supports any sampler and noise level* as a core advantage over DDPO. However,  no evidence is provided to substantiate this claim, as all experiments use only Euler-Maruyama sampling.


2. The paper's claim of consistency with *policy gradient theory* is questionable. The authors use a uniform weight
$w(t)=1$  instead of the correct ELBO weight, introducing bias into the importance-weighted ratio. This results in a biased estimator of the true likelihood ratio, violating the foundation of policy gradient methods. While the paper acknowledges this discrepancy, but not theoretically addressed.


3. The paper seems to claim that AWM is a lower-variance version of DDPO by representing formulation, but they optimize different objectives: DDPO uses a per-step policy with temporal credit, as the authors convey, while AWM uses a sequence policy without temporal credit. This isn't variance reduction but rather a fundamentally different approach. The theoretical contribution remains unclear, as the reported 24× speedup lacks a breakdown into: (1) computational cost (1 loss vs. T likelihoods), (2) variance reduction, and (3) implementation differences (e.g., LLaDA 1.5 tricks mentioned on line 310 but not ablated).


4.  The paper's presentation significantly hinders a clear understanding of the methodology. It not only fails to provide a clear comparison of the loss function for the Flow-GRPO baseline, but it also fails to clearly present the loss objective for AWM.

5.  The paper's notation and logic are flawed and difficult to follow. For example, although the paper claims to select a flow-matching noise schedule in Sec. 2.1, the manuscript repeatedly switches between $v_\theta$, $s_\theta$, and $D_\theta$.  Some notations appear rigidly without any explanation, such as  $\pi_{\theta_{\text{old}}}$ and $v_{\theta_{\text{ref}}}$.  Additionally, some formulas are unclear, such as the awkward appearance of Eq. (8).

**Questions:**

1.  Why does the paper lack comparisons to any preference alignment baselines, aside from Flow-GRPO?

2.   Does the proposed method tend to sacrifice generation diversity for high rewards more than Flow-GRPO's per-step strategy?

3.  Some claims are vague and misleading. For example, the paper states, "Euler–Maruyama can increase estimator variance, and using an exponential integrator yields the *analytical variance*.  Is this conclusion valid under the same conditions?

---

> ### Author Response · Authors · 2025-11-24
>
> **(Weakness 1):** decoupling from sampling ; all results use Euler–Maruyama.
>
> **(Response 1):** We thank the reviewer for this comment, as it allows us to clarify a **fundamental design advantage** of AWM.
>
> The reviewer’s confusion is understandable, as such decoupling is impossible for DDPO-based methods. However, AWM's "sampler-agnostic" nature is not just an empirical claim to be proven; it is a structural property of our algorithm's design.
>
> AWM's sampling and training is natually decoupled by design: Specifically, the AWM objective function (Algorithm 1, line 12) operates **only** on the **final generated sample $x_0$** and a $x_t$ obtained via **the forward noising process**. In contrast, DDPO defines its loss on sampler-dependent intermediate steps $\log p(x_{t-1}|x_t)$, , which tightly couples it to the specifics of the sampling trajectory. AWM's loss function, by design, has **zero dependence** on the **sampling path** or any **intermediate states**, making it **inherently sampler-agnostic**.
>
> We agree that validating across diverse samplers is important. However, our focus in this work is to isolate **the effect of the training algorithm itself**, and we therefore used the same Euler–Maruyama sampler as the flow-GRPO baseline to ensure fair comparison in all experiments.
>
> To further validate the sampler-agnostic claim, we conducted additional experiments using both **first-order ODE** and **first-order SDE** samplers on the **OCR** and **ImageReward** merics. As shown below (see Appendix F for details), AWM demonstrates consistent trends in reward improvement across both sampling methods:
>
> **OCR**
> | Steps | 0| 50 | 100 | 150 | 200 | 250 |
> |-|-|-|-|-|-|-|
> | Euler (ODE) |0.55| 0.84 | 0.90 | 0.93 | 0.95 | 0.96|
> | Euler-Maruyama (SDE) |0.53| 0.88 | 0.91 | 0.93 | 0.95 | 0.95|
>
> **Imagereward**
> | Steps | 0| 50 | 100 | 150 | 200 | 250 |
> |-|-|-|-|-|-|-|
> | Euler (ODE) |1.04 |1.57 |1.57|1.67|1.69|1.74|
> | Euler-Maruyama (SDE) |1.06| 1.50| 1.64| 1.67| 1.63| 1.74|
>
> We report a 10-step moving average to reduce local fluctuations. These results confirm that AWM’s performance is robust across different samplers, aligning with its timestep-decoupled and sampler-independent formulation.
>
> ---
>
> **(Weakness 2):** Uniform weight replaces the correct ELBO weight, biasing the importance-weighted likelihood ratio and undermining policy-gradient validity; discrepancy not theoretically resolved.
>
>
> **(Response 2):** We appreciate the reviewer raising this important point regarding the consistency. We would like to clarify this design choice from two perspectives:
>
> **1. Alignment with Established Pretraining Practices**
> As established in foundational works [1, 2, 3], while the ELBO weighting theoretically optimizes log-likelihood, it **allocates excessive capacity to high-frequency** (small $t$), leading to **suboptimal visual quality and FID scores**. Consequently, more uniform weighting schemes has become the **de facto standard for pretraining** to balance visual quality across all noise levels.
>
> **2. Invariance of the Optimal Solution**
> Theoretically, the weighting function $w(t)$ serves as a task-balancing mechanism. As noted in [4], if the model has sufficient capacity (or if parameters are not shared across t in the model), the **global optimum** of the score matching objective is invariant to the choice of $w(t)$. Therefore, changing the weight alters the optimization trajectory and convergence speed, especially in the current weight sharing case, but does not theoretically change the optimal objective.
>
> **References:**
>
> [1] Denoising Diffusion Probabilistic Models. Jonathan Ho, etc. 2020.
>
> [2] Improved Denoising Diffusion Probabilistic Models. Alex Nichol, etc. 2020.
>
> [3] Maximum Likelihood Training of Score-Based Diffusion Models. Yang Song, etc. 2021.
>
> [4] Denoising Diffusion Implicit Models. Jiaming Song, etc. 2022.

---

> > ### Author Response · Authors · 2025-11-24
> >
> > **(Weakness 3):** AWM’s sequence-level policy departs from DDPO’s temporally-credited per-step policy; not merely variance reduction. Theoretical novelty and the claimed 24× speedup remain unclear.
> >
> >
> > **(Response 3):** We would like to clarify the following points:
> >
> > **1. View on Temporal Credit.**
> > We respectfully disagree with the statement that *“this isn’t variance reduction but rather a fundamentally different approach.”* In the context of reinforcement learning (RL) post-training for large language models (LLMs), both token-level and sequence-level perspectives coexist—analogous to GRPO and GSPO. However, in the **on-policy** case, when equipped with **outcome-based reward**, these two perspectives (token-level vs. sequence-level) converge to yield **the same algorithm[5]**. In both Flow-GRPO and our method, all experiments adopt outcome-based rewards, where the reward depends solely on the clean final output $x_0$. Crucially, since rewards are derived solely from the final outcome ($x_0$), **Flow-GRPO assigns the identical advantage value to every timestep within a trajectory**. Consequently, it effectively operates **without temporal credit assignment**.
> >
> > **2. Necessity of the LLaDA 1.5 Trick.**
> > The LLaDA 1.5 trick is crucial for maintaining the estimated likelihood ratio
> > $\frac{\pi_{\theta}}{\pi_{\theta_{old}}} = 1$
> > in the on-policy case
> > ($\pi_\theta = \pi_{\theta_{old}}$).
> >
> > This mechanism is also **implicitly implemented** in nearly all diffusion-RL baseline, including **Flow-GRPO, diffusion-DPO**, etc.
> >
> > **3. Detailed Analysis of Computational Efficiency.**
> > To further demonstrate the efficiency of our method, we benchmarked AWM on the ImageReward (SD3.5M) dataset. The table below presents the 10-step moving average of rewards:
> >
> > | Steps | 0| 40 | 80 | 120 | 160 | 200 | 240 | 280 | 320 | 360 |
> > |-|-|-|-|-|-|-|-|-|-|-|
> > | Reward AWM(Ours) |1.06| 1.51 | 1.55 |1.67|1.68|1.69|1.68|1.68|1.72|1.75|
> > | Reward Flow-GRPO |1.06| 1.10 | 1.23 |1.30|1.35|1.41|1.46|1.49| 1.52| 1.52|
> >
> >
> > **Efficiency Breakdown:**:
> >
> > * **Gradient-update efficiency**: AWM achieves **7× fewer update steps** to reach similar reward (40 vs. 280 steps).
> >
> > * **Per-step GPU time**: AWM is **2.75× faster per step** (283s vs. 781s). This is because Flow-GRPO requires classifier-free guidance (CFG) during training to match its sampling distribution, whereas AWM avoids CFG by aligning with pretraining.
> >
> > * **Total GPU hours**: Combined, these lead to an overall **19.32× speedup**.
> >
> > **4. Theoretical and Empirical Support for Variance Reduction.**
> > Our variance reduction claim is not based solely on theoretical analysis. We also conducted controlled pretraining experiments on CIFAR-10 and ImageNet-64 (See Figure 3). In both cases, the pretraining objective corresponding to AWM exhibits significantly faster and more stable convergence compared to DDPO-style objectives. These findings offer direct empirical support for the variance reduction interpretation.
> >
> > While we acknowledge that alternative explanations may also account for the observed speedup, the empirical gains in both efficiency and stability achieved by AWM remain consistent across all evaluated settings.
> >
> > [5] Group Sequence Policy Optimization. Chujie Zheng, etc. 2025.
> >
> > ---
> >
> > **(Weakness 4):** Presentation hampers understanding of core methodology, including Flow-GRPO loss and AWM objective.
> >
> > **(Response 4):** We appreciate the reviewer’s feedback regarding the clarity of the methodology. In the revised manuscript, we have reorganized the methodology section to explicitly define and locate the training objectives for both the baseline and our proposed method:
> >
> > *   **AWM Objective:** The AWM training objective is now clearly defined in **Line 296 and Equation 9**. This formulation integrates the GRPO framework with a likelihood ratio estimated via differences in Flow Matching losses **(Equation 11)**.
> > *   **Flow-GRPO (DDPO) Objective:** As Flow-GRPO follows the standard DDPO formulation, we have included a detailed derivation and explicit definition of its per-step Gaussian likelihood objective in **Appendix D**.
> >
> > To further clarify the methodological differences, we summarize key distinctions between Flow-GRPO and AWM across three core dimensions, including sampling, reward, and training, in the table below:
> >
> > | Aspect | Flow-GRPO (Baseline) | AWM (Ours) |
> > | :--- | :--- | :--- |
> > | **Sampling** | **Coupled:** Requires Euler-Maruyama discretization. | **Decoupled:** Supports arbitrary ODE/SDE samplers as our algorithm only relies on $x_0$. |
> > | **Reward** | **Outcome-based:** Evaluated solely on the final generated clean image $x_0$. | **Outcome-based:** Evaluated solely on the final generated clean image $x_0$. |
> > | **Training** | Optimizes the advantage weighted Gaussian likelihood of the noisy next step $x_{t-1}$ given $x_t$. | Optimizes the advantage-weighted Flow Matching loss on clean data $x_0$ at some $t$. |

---

> > > ### Author Response · Authors · 2025-11-24
> > >
> > > **(Weakness 5):** Difficult-to-follow notation (switching between velocity/score/$x_0$ prediction) and undefined $\pi_{ref}$/$\pi_{old}$; Eq. (8) lacks clarity.
> > >
> > > **(Response 5):** We apologize for the lack of clarity in the original submission. In the revised manuscript, we have made several improvements to enhance the logical flow and interpretability:
> > >
> > > **1. Comprehensive Notation Table (Appendix A).**
> > > To address ambiguity in notation, we have added **a detailed Notation Table in Appendix A (Table 4)**, that explicitly defines all symbols related to diffusion processes ($x_t, \epsilon$), RL components ($\pi_{\text{old}}, r, A$), and the different network parameterizations (velocity, score, data).
> > >
> > > **2. Rationale for Switching Parameterizations.**
> > > We respectfully clarify that the use of different parameterizations is not a flaw but a necessity to align with (1) theory, (2) modern architecture standards, and (3) baseline implementation practices.
> > > Specifically:
> > > *   **Score ($s_{\theta}$):** Used in **Sec. 2** for theoretical analysis, as it is the standard mathematical language for proving the equivalence between DDPO and DSM.
> > > *   **Velocity ($v_{\theta}$):** Used in **Sec. 3**, as our primary backbones (SD3.5M, FLUX) follow the Flow Matching paradigm, which inherently predicts velocity.
> > > *   **Clean Data Prediction ($D_{\theta}$):** Used in **Eq. (8)** for evaluation on the EDM (Karras et al., 2022), which is based on $x_0$-prediction.
> > >
> > >
> > > **3. Clarification on $\pi_{\text{old}}$ and $\pi_{\text{ref}}$.**
> > > To eliminate ambiguity, we have added explicit definitions in **Section 3, Line 296**. Specifically:
> > > *   $\pi_{\theta_{\texttt{old}}}$: The policy used to collect sampling data, used for computing importance sampling ratios.
> > > *   $\pi_{\theta_{\texttt{ref}}}$: The frozen reference model (usually the pretrained base model), used to calculate the KL divergence penalty.
> > >
> > > **4. Derivation of Equation (8) and Definition of $D_{\theta}$.**
> > > We now provide the definition of $D_\theta$ directly before Equation (7). This equation expresses the Noisy DSM objective adapted to the EDM setting, where the standard $x_0$ target is replaced by one derived from noisy score matching.
> > >
> > > A complete step-by-step derivation is now included in Appendix C.4 to make this formulation fully transparent.
> > >
> > > ---
> > >
> > > **(Question 1):** Why does the paper lack comparisons to any preference alignment baselines, aside from Flow-GRPO?
> > >
> > >
> > > **(Response 1):** Below, we first clarify why certain comparisons are not applicable, and then present additional empirical comparisons with two strong and relevant baselines.
> > >
> > > **1. Inapplicability of Certain Baselines**
> > > As discussed in Section E of the appendix (Related Work), some prior methods are incompatible with our evaluation setup due to fundamental assumptions about reward differentiability. For examples, **ReFL (Reward Fine-Tuning)** and **DRaFT** both require *differentiable reward functions* to backpropagate gradients. However, our evaluation benchmarks (GenEval, OCR) involve *non-differentiable* metrics, where such reward backpropagation is ***inherently*** not applicable. In contrast, AWM optimizes *sampled, non-differentiable* rewards via a policy-gradient formulation, making it suitable for these domains.
> > >
> > > **2. Additional Comparisons Newly Include**
> > >
> > > To further strengthen our evaluation, we have added comparisons to two recent and highly relevant diffusion-RL baselines:
> > >
> > > - **Compared with Dance-GRPO on FLUX**
> > >
> > > | Reward | Pick Score | GenEval |
> > > |-|-|-|
> > > | Dance-GRPO |23.0 | 0.687 |
> > > | AWM (Ours) |**23.2**| **0.938** |
> > >
> > > - **Compared with DiffusionNFT on SD3.5-M**
> > >
> > > | Reward | OCR | ImageReward|
> > > |-|-|-|
> > > | DiffusionNFT |0.91 | 1.49|
> > > | AWM (Ours) |**0.95**| **1.75**|
> > >
> > > These results demonstrate that AWM consistently outperforms prior methods across diverse models and reward types, underscoring its complementary advantages in diffusion-based RL settings.

---

> > > > ### Author Response · Authors · 2025-11-24
> > > >
> > > > **(Question 2):** Does the proposed method tend to sacrifice generation diversity for high rewards more than Flow-GRPO's per-step strategy?
> > > >
> > > > **(Response 2):** We additionally checked whether AWM sacrifices diversity by running multiple seeds on a random subset of prompts using the AWM-finetuned FLUX model. The generated images show clear variation in both content and style across seeds, confirming that our method does not compromise generation diversity. The results have been updated in Appendix G.
> > > >
> > > > ---
> > > >
> > > > **(Question 3)** Vague claim that EM increases estimator variance while exponential integrators yield analytical variance; unclear if both results hold under comparable conditions.
> > > >
> > > > **(Response 3):** We clarify that this conclusion is valid and stems from the discretization error of the diffusion term in the reverse-time process.
> > > >
> > > > Consider the diffusion coefficient $g(t) = \sqrt{\frac{2t}{1-t}}$ in the reverse SDE. The reverse process solves backwards from $t$ to $t - \Delta t$.
> > > > 1.  **Monotonicity:** $g(t)$ is strictly increasing with respect to $t$. Therefore, for any time $s$ in the interval $[t-\Delta t, t)$, $g(s) < g(t)$.
> > > > 2.  **Euler-Maruyama (EM) Variance:** EM approximates the process using the coefficient at the starting time $t$. The variance injected in one step is:
> > > >     $$ \sigma^2_{\text{EM}} = g(t)^2 \Delta t $$
> > > > 3.  **Analytical Variance:** The exact variance accumulated over the interval is the integral of the instantaneous variance:
> > > >     $$ \sigma^2_{\text{Exact}} = \int_{t-\Delta t}^{t} g(s)^2 ds $$
> > > >
> > > > **Conclusion:** Since $g(t)$ is the maximum value within the interval $[t-\Delta t, t]$, it strictly follows that:
> > > > $$ \int_{t-\Delta t}^{t} g(s)^2 ds < g(t)^2 \Delta t \implies \sigma^2_{\text{Exact}} < \sigma^2_{\text{EM}} $$
> > > > Thus, Euler-Maruyama systematically overestimates the diffusion noise, leading to higher estimator variance compared to the analytical form.

---

### Official Review · Reviewer_abUX · 2025-11-01

**Soundness:** 2
**Presentation:** 3
**Contribution:** 3
**Rating:** 4
**Confidence:** 3

**Summary:**

The paper identifies a key misalignment between RL and pretraining in diffusion models, unlike in the field of LLMs where both stages share a consistent log-likelihood objective. This mismatch leads to high variance and slow convergence. The authors theoretically prove that DDPO is equivalent to performing DSM with noisy data and that this noisy conditioning inflates variance. To resolve this, they propose Advantage-Weighted Matching (AWM), which applies advantage weighted score or flow matching to preserve the pretraining objective while amplifying high-reward samples.

**Strengths:**

1.	The proof that DDPO is equivalent to noisy DSM spells out why existing RL methods for diffusion models suffer from high variance and slow convergence. This analysis is important and fills a gap in understanding diffusion RL.
2.	AWM reuses the same DSM/FM objective as diffusion pretraining and adds advantage based weights. This symmetry makes RL for diffusion conceptually similar to RL for LLMs and allows training and sampling to be decoupled, enabling flexible samplers.
3.	AWM reduces the variance of policy gradients and achieves significant speed ups in experiments without sacrificing quality. The speed up is particularly important for computationally intensive diffusion models.

**Weaknesses:**

1.	The DDPO to DSM equivalence theorem relies on the time‑reversal property of diffusion processes and explicitly ignores the Euler–Maruyama discretization error. This assumption simplifies the proof but is not generally valid during practical training, where finite step sizes and non‑Gaussian noise schedules can significantly affect the reverse‑process likelihood. So, this equivalence may break down in real implementations.
2.	There is a potential for weight collapse. Especially when rewards have high variance, advantage weighting might disproportionately emphasize a small subset of samples, potentially harming diversity. The paper does not discuss strategies to prevent such weight collapse or to normalize advantages.
3.	AWM is positioned as sampler-agnostic and timestep-decoupled, but the experiments predominantly use a single sampler. Validating with diverse ODE or SDE samplers and few-step distilled samplers would substantiate the claimed flexibility.
4.	The paper mentions ReFL, DRaFT and FMPG but does not provide direct empirical or analytical comparisons with them. Since these methods represent major alternative directions for diffusion RL, i.e., reward backpropagation and ELBO-based policy gradients, a deeper comparison would stand out AWM’s complementary strengths, such as variance reduction and sampler-agnostic training. Only comparing against Flow-GRPO is limited.

**Questions:**

1. How does AWM’s performance depend on the noise and bias in the reward function? For example, would inaccurate or sparse rewards lead to unstable advantage weights?

2. The paper evaluates SD3.5M and Flux. How does AWM scale to larger models (e.g., SD‑XL) or to diffusion models for video, audio, or multimodal tasks?

---

> ### Author Response · Authors · 2025-11-24
>
> **(Weakness 1):** The DDPO to DSM equivalence theorem relies on the time‑reversal property of diffusion processes and explicitly ignores the Euler–Maruyama discretization error. Practical diffusion implementations may violate assumptions.
>
> **(Response 1):** We thank the reviewer for the comments, which allow us to clarify two critical points.
>
> **1. On discretization error:**
> This is a flaw of DDPO, not a simplification in our analysis. Instead, AWM mitigates the discretization error through the forward process.**
>
> The reviewer correctly notes that the equivalence in Theorem 1 holds up to the discretization error. We emphasize that this is not a simplification in our analysis, but rather underscores a **core, inherent flaw of the DDPO algorithm itself.**
>
> * **DDPO's objective relies on discretization:** The DDPO objective is, by definition, the log-likelihood of a *specific, discrete, reverse-time transition* $\log p(x_{t-1}|x_t)$. This means DDPO's training objective is tied to the discretization, as $x_{t-1}|x_t$ approximates Gaussian only when the time gap is small[1].
> * **Role of the forward process in AWM:** In contrast, the forward transition $x_t \mid x_{t-1}$ is defined analytically and therefore does not incur numerical discretization error. In Theorem 1, the only approximation arises when $(x_{t-1}, x_t)$ is simulated via the Euler–Maruyama discretization of the reverse process; drawing $(x_{t-1}, x_t)$ from the forward process remains exact. This is the sole source of discrepancy in Theorem 1.
> * **Conclusion:** Thus, the "*discretization error*" is **not an analytical oversight** but **an inherent additional error present in DDPO** and avoided by AWM. The faster convergence of AWM is, in part, due to it avoids the discretization error that burdens DDPO.
>
> **2. On Gaussian assumption**
>
> Gaussian diffusion and Gaussian flow matching are de-facto frameworks, our method focus on this domain. While extending to non-Gaussian flow matching is an interesting direction for future work, it falls outside the scope of the current study.
>
> [1] Tackling the Generative Learning Trilemma with Denoising Diffusion GANs. Zhisheng Xiao, etc. 2022.
>
> ---
>
> **(Weakness 2):** Risk of weight collapse under high-variance rewards; "missing normalization strategy".
>
> **(Response 2):** We thank the reviewer for raising this important concern. The potential instability or "weight collapse" caused by high-variance rewards is a well-known challenge in policy gradient methods. To address this, we adopt two common techniques to prevent the potential collapse.
>
> 1. **KL Regularization** (refer to Sec. 3, Line 301, Equation 9, and Line 319, Equation 12): To prevent "weight collapse" and encourage diversity, our objective function incorporates a KL regularization term. This term constrains the policy, keeping it close to the original pretrained reference model. It explicitly penalizes the policy for deviating too far, thus preventing it from over-optimizing and "collapsing" onto a small subset of high-reward samples.
>
> 2. **Advantage Normalization** (ref to Sec. 3 Line 293): We explicitly stated in the original manuscript that we compute advantages using the "group relative mean" approach from GRPO. Specifically, we center the rewards within each batch by subtracting the batch mean, and then normalize by the global standard deviation. This stabilizes training by reducing variance. We have further clarified the advantage computation in Line 293 in the updated version.
>
> We additionally checked whether AWM sacrifices diversity by running multiple seeds on a random subset of prompts using the AWM-finetuned FLUX model. The generated images show clear variation in both content and style across seeds, confirming that our method does not compromise generation diversity. The results have been updated in Appendix G.

---

> > ### Author Response · Authors · 2025-11-24
> >
> > **(Weakness 3):** AWM claimed sampler-agnostic and timestep-decoupled, yet tested with one sampler only.
> >
> > **(Response 3):** We appreciate the reviewer’s suggestion. We clarify that sampler-agnosticism is a property that holds by construction in AWM, not an empirical claim that depends on specific samplers used in experiments. Specifically, the AWM objective function (Algorithm 1, line 12) operates **only** on the **final generated sample $x_0$** and a $x_t$ obtained via forward noising. Unlike DDPO, which defines its loss on sampler-dependent intermediate steps $\log p(x_{t-1}|x_t)$ , our loss function has **zero dependence** on the **sampling path** or any **intermediate states**. Therefore, AWM is **inherently sampler-agnostic** and timestep-decoupled.
> >
> > We agree that validating across diverse samplers is important. Our goal in this work is to isolate **the effect of the training algorithm itself**, keeping the sampling procedure identical to the flow-GRPO baseline for **fair comparisons**. We therefore used the same Euler–Maruyama sampler across all experiments.
> >
> > To further verify sampler-agnostic behavior, we evaluated AWM using **first-order ODE**, and **first-order SDE** samplers on **OCR and Imagereward**. In both cases, we observe consistent trends in reward improvement, as summarized below (full results have been updated provided in Appendix F), where a 10-step moving average is reported to smooth local fluctuations.
> >
> >
> > **OCR**
> > | Steps | 0| 50 | 100 | 150 | 200 | 250 |
> > |-|-|-|-|-|-|-|
> > | Euler (ODE) |0.55| 0.84 | 0.90 | 0.93 | 0.95 | 0.96|
> > | Euler-Maruyama (SDE) |0.53| 0.88 | 0.91 | 0.93 | 0.95 | 0.95|
> >
> > **Imagereward**
> > | Steps | 0| 50 | 100 | 150 | 200 | 250 |
> > |-|-|-|-|-|-|-|
> > | Euler (ODE) |1.04 |1.57 |1.57|1.67|1.69|1.74|
> > | Euler-Maruyama (SDE) |1.06| 1.50| 1.64| 1.67| 1.63| 1.74|
> >
> >
> > While designing step-distilled or sampler-adaptive methods is a promising direction, it is orthogonal to our focus. Since our formulation is independent of the sampler’s discretization, it **naturally supports** integration with such techniques and can be extended accordingly in future work.
> >
> > ---
> >
> > **(Weakness 4):** No direct comparison with reward-backpropagation and ELBO-gradient baselines (ReFL/DRaFT/FMPG); limits demonstration of AWM’s advantages.
> >
> > **(Response 4):** We thank the reviewer for the thoughtful question. Below, we first clarify why certain comparisons are not applicable, and then present additional empirical comparisons with two strong and relevant baselines.
> >
> > **1. Inapplicability of Certain Baselines**
> > We have clarified in the paper (Sec E in the appendix, related work) that
> >
> > - **ReFL (Reward Fine-Tuning)** and **DRaFT** both require *differentiable reward functions* to backpropagate gradients. Hoeveer, our evaluation benchmarks (GenEval, OCR) involve *non-differentiable* metrics, where such reward backpropagation is ***inherently*** not applicable. In contrast, AWM optimizes *sampled, non-differentiable* rewards via a policy-gradient formulation, making it suitable for these domains.
> > - **FMPG (Flow-Matching Policy Gradient)** is concurrent to our work and algorithmically related. However, FMPG does **not analyze the connection to DDPO**, nor does it evaluate on text-to-image diffusion tasks. Instead, its focus remains on conventional RL environments.
> >
> > **2. Additional comparisons newly include**
> >
> > Despite the limitations above, we strengthened the empirical section by adding comparisons to two strong and highly relevant diffusion-RL baselines:
> >
> > **Compared with Dance-GRPO on FLUX**
> > | Reward | Pick Score | GenEval |
> > |-|-|-|
> > | Dance-GRPO[2] |23.0 | 0.687 |
> > | AWM (Ours) |**23.2**| **0.938** |
> >
> > **Compared with DiffusionNFT on SD3.5-M**
> > | Reward | OCR | ImageReward
> > |-|-|-|
> > | DiffusionNFT[3] |0.91 | 1.49|
> > | AWM (Ours) |**0.95**|  **1.75**|
> >
> > These results demonstrate that AWM consistently outperforms prior methods across diverse models and reward types, underscoring its complementary advantages in diffusion-based RL settings.
> >
> >
> > [2] DanceGRPO: Unleashing GRPO on Visual Generation
> >
> > [3] DiffusionNFT: Online Diffusion Reinforcement with Forward Process

---

> > > ### Author Response · Authors · 2025-11-24
> > >
> > > **(Question 1)** How reward noise/bias affects AWM stability and advantage-weight reliability.
> > >
> > > **(Response 1)** We thank the reviewer for raising this critical point. This is a fundamental challenge for all preference-based reinforcement learning algorithms, as the quality of the reward function (its noise, bias, and sparsity) dictates the upper bound of alignment for any method. This is a shared limitation for both AWM and all other RL algorithms.
> > >
> > > Our contribution lies in developing an algorithm, i.e., AWM, that operates under the same noisy, biased, and imperfect reward signals as existing baselines, yet achieves significantly greater stability and efficiency. In particular, through the combination of GRPO-based advantage normalization and KL regularization, AWM achieves an 8.02× speedup on GenEval, demonstrating improved sample efficiency despite the inherent limitations of the reward signal.
> > >
> > > ---
> > >
> > > **(Question 2):** Applicability of AWM to larger diffusion architectures and non-image domains.
> > >
> > > **(Response 2):** We thank the reviewer for this question regarding scaling and generalization. We believe our experiments already demonstrate robust scaling. While **SD-XL (2.6B parameters)** is a valid benchmark, it is similar in scale to our tested **SD3.5-M (2.5B parameters)** and substantially smaller than **FLUX (12B parameters)**. Our experiments on the 12B parameter FLUX model show that AWM **scales effectively** to a model that is **significantly larger (over 4x) than SD-XL**.
> > >
> > > We chose SD3.5M and FLUX not only for their scale but because they represent the current state-of-the-art, both being **modern architectures**. We felt it was more impactful to demonstrate AWM's effectiveness on these newer architectures rather than on the older SD-XL.
> > >
> > > For other Modalities, we agree that extending AWM to video, audio, and other multimodal tasks is an exciting and important future direction. However, to ensure a focused and thorough investigation and given pratical constraints on computational resources, we limited the scope of this initial study to text-to-image setting.

---

### Official Review · Reviewer_pVy6 · 2025-11-01

**Soundness:** 3
**Presentation:** 3
**Contribution:** 3
**Rating:** 6
**Confidence:** 4

**Summary:**

This paper introduces and studies an diffusion RL post-training approach -- Advantage Weighted Matching (AWM). By reweighting each sample by its advantage, this approaches achieves low variance and faster convergence. Experiments have been conducted on Stable diffusion 3.5 and Flux.

**Strengths:**

The paper is well-written, and I mostly enjoyed reading it. The idea is clear: reweighting the sample by it advantage relying on the GRPO pipeline. The numerical experiments also support the proposed approach.

**Weaknesses:**

There are several weaknesses that I hope the authors can address and explain.

(1) It seems to me that incorporating the advantage in policy gradient is natural (line 316), and this idea has been explored elsewhere. The authors may explain what is the main novelty here.

(2) Lemma 1: this result is well known (see e.g., Score-based diffusion models via stochastic differential equations, Statist. Surv., Tang and Zhao, Thm 4.3), and in the original paper of Vincent. The authors may simply refer to there.

(3) Thm 1: the viewpoint that DDPO is doing denoising score matching does not seem to be a "secret". See e.g.,
Reward-Directed Score-Based Diffusion Models via q-Learning, arXiv:2409.04832, Gao, Zha and Zhou;
Score as Action: Fine-Tuning Diffusion Generative Models by Continuous-time Reinforcement Learning, ICML25, Zhao et al.

(4) Experiments: the authors did experiments using the PickScore. I wonder if the results are similar in other metrics, e.g., HPSv2, Vendi score and ImageReward?

(5) Figure 7(b): it is interesting that for beta = 0.2, the training curve goes done rapidly; while for beta = 0.4 it works well. I wonder if there is a "threshold" (between 0.2 and 0.4) below which the curve fails to improve.

**Questions:**

See weakness.

---

> ### Author Response · Authors · 2025-11-24
>
> **(Weakness 1):** Use of advantage functions is standard; clarify novelty.
>
> **(Response 1):** We agree that using advantages in policy gradients is standard and not our contribution. **However, our novelty is orthogonal to the choice of advantage**: the key contribution lies in how the policy likelihood $\pi_\theta$ is defined for diffusion models in the specific domain of diffusion post-training.
>
> Rather than using the *per-step* conditional likelihood $\pi_\theta(x_{t-1}\mid x_t, c)$ as in DDPO, we formulate the policy as ***the sequence-level likelihood*** $\pi_\theta(x_0\mid c)$ induced by the entire diffusion process, and we optimize its ELBO/score-matching surrogate accordingly. This formulation more faithfully captures the distribution that ultimately defines the generated sample.
>
> Moreover, on the theoretical side, we show that **DDPO's objective is equivalent to denoising score matching with noisy targets $x_{t-\Delta t}$**, which inflates target variance and slows optimization (see Sec. 2.3). Our formulation avoids the variance inflation and leads to more stable and efficient training.
>
> ---
>
> **(Weakness 2):** Well-known Lemma 1; should reference established results.
>
> **(Response 2):** Thank you for sharing the related work. We have carefully reviewed the suggested reference and would like to offer the following clarification. The classical denoising score-matching (DSM) identities (e.g., Vincent, 2011[1]; Tang & Zhao, 2024[2]) indeed correspond to the **special case $s=0$** in our Lemma 1. Our contribution is a strict generalization of these results: we provides a further **extension** from clean target $x_0$ to noisy target $x_s$, which is essential for our diffusion post-training setting. That is why we write "This lemma **generalizes** the well-known result" (Line 215). We have updated the manuscript to explicitly cite both papers and clarified this connection.
>
> [1] A Connection Between Score Matching and Denoising Autoencoders
>
> [2] Score-based Diffusion Models via Stochastic Differential Equations -- a Technical Tutorial
>
>
> ---
>
> **(Weakness 3):** Thm. 1 reflects a known DDPO–score-matching connection.
>
> **(Response 3):** We carefully checked both references suggested by the reviewer and respectfully disagree that Theorem 1 appears in either work:
>
> - Gao, Zha & Zhou (arXiv:2409.04832)[3]
> - Zhao et al. (ICML’25)[4]
>
> Moreover, Gao et al. explicitly state that their framework and the theoretical results are *“fundamentally different”* from DDPO/DPOK, underscoring that they do not analyze DDPO.
>
> To our knowledge, neither paper provides a theorem or lemma establishing the equivalence we prove, namely, **DDPO ≡ DSM conditioned on noisy intermediates**.
>
> If we are overlooking a specific statement or derivation in these works, we would appreciate any further comments on which specific parts of the two referenced papers the reviewer finds related to Theorem 1. We would be more than happy to provide additional clarification if needed during the paper discussion phase. **For completeness, we have included both works in the related work section E.4 (Line 1125) in the updated version.**
>
> [3] Reward-Directed Score-Based Diffusion Models via q-Learning
>
> [4] Score as Action: Fine-Tuning Diffusion Generative Models by Continuous-time Reinforcement Learning
>
> ---
>
> **(Weakness 4):** Evaluation limited to PickScore; missing other metrics.
>
> **(Response 4):** We thank the reviewer for the suggestion. We additionally evaluate our method on **ImageReward** for SD3.5M, a widely adopted reward model. The results are shown below:
>
> | Steps | 0| 40 | 80 | 120 | 160 | 200 | 240 | 280 | 320 | 360 |
> |-|-|-|-|-|-|-|-|-|-|-|
> | Reward AWM(Ours) |1.06| 1.51 | 1.55 |1.67|1.68|1.69|1.68|1.68|1.72|1.75|
> | Reward Flow-GRPO |1.06| 1.10 | 1.23 |1.30|1.35|1.41|1.46|1.49| 1.52| 1.52|
>
> We report a 10-step moving average to reduce local fluctuations. As shown, our method consistently achieves significantly higher reward across training steps.
>
> As for efficiency evaluation, under identical sampling steps and batch size, AWM achieves a **7× acceleration in terms of gradient update steps** (reaching similar ImageReward improvements at 40 vs. 280 steps). In terms of per-step GPU time, AWM is **2.75× faster** (283 s vs. 781 s), because Flow-GRPO requires CFG during training to match the sampling trajectory, whereas our method does not require CFG (consistent with pretraining).
>
> **Overall**, when combining step efficiency and per-step computational savings, **AWM achieves a total of 19.32× reduction in GPU hours** compared to Flow-GRPO on ImageReward. We have updated these results in the Appendix F for comprehensive evaluation.

---

> > ### Author Response · Authors · 2025-11-24
> >
> > **(Weakness 5):** Unclear beta threshold for stable training.
> >
> > **(Response 5):** We thank the reviewer for the insightful question. Following the suggestion, we have concluded an additional evaluation of our method using an intermediate value $\beta = 0.3$. The results are shown below:
> >
> > | Steps | 0| 40 | 80 | 120 | 160 |  240 |
> > |-|-|-|-|-|-|-|
> > | Geneval AWM $\beta=0.3$ |0.63|0.83| 0.86 |0.86|0.91| 0.92|
> >
> > We would like to clarify that the training instability in reinforcement learning (RL) is a well-known issue, including in RLHF for LLMs. There is typically no sharp or universal threshold separating stable and unstable regimes. Therefore, while $\beta = 0.2$ leads to collapse and $\beta = 0.4$ remains stable, this should not be interpreted as a strict boundaty. Instead, training stability tends to improve progressively with increasing $\beta$.

---

### Author Response · Authors · 2025-11-24
**Response to All**

Dear All reviewers, former AC, and new AC:


We sincerely appreciate all reviewers for the time and efforts to review our paper.
We are glad to find that reviewers recognized the following contributions of our work:

-   **Strong empirical performance and substantial speed-ups:** All reviewers (pVy6, abUX, ZC3a, FSZB) highlighted our efficiency gains without compromising generation quality. Specifically, AWM achieves up to **24× faster training** than Flow-GRPO on **SD-3.5-M and FLUX**.

-   **Novel theoretical analysis linking DDPO and Noisy DSM:** Reviewers (abUX, FSZB) valued our proof that DDPO is theoretically equivalent to **Denoising Score Matching (DSM) with noisy targets**. This crucial insight explains the high variance and slow convergence of prior methods compared to AWM, which aligns RL post-training with the lower-variance **clean-data pretraining objective**.

-   **Clear writing and presentation quality:** Reviewers (pVy6, FSZB) commended the clarity of our derivations and the exposition of the proposed Advantage Weighted Matching (AWM) pipeline.



We also thank all reviewers for their insightful and constructive suggestions, which help further improve our paper. In addition to the pointwise responses below, we summarize the major revision in the rebuttal according to reviewers’ suggestions.

* **Clarification of Novelty and Relation to Prior RWR-style Methods** ([FSZB])
We explicitly contrasted AWM with existing reward-weighted regression, emphasizing the use of **possibly negative advantages** as weight instead of strictly positive weights.

* **ELBO Weighting vs. Uniform Time Weighting** ([ZC3a, FSZB])
  We added a more detailed discussion on the mismatch between the theoretical ELBO weighting and the practically adopted uniform weighting in modern diffusion pretraining. We explain why uniform weighting has become the de facto standard (better visual quality and balanced noise levels) and clarify that, under sufficient capacity, **changing the weighting** affects optimization dynamics but **not the optimal solution**.

* **Discretization Error in Theorem 1 and Its Implications for DDPO** ([abUX, FSZB])
  We clarified that the discretization term in Theorem 1 is not a missing detail in our analysis, but an inherent **limitation of DDPO** itself, whose objective is tied to the discrete reverse transition $x_{t-1} \mid x_t$.

* **Sampler-agnostic and Timestep-decoupled Nature of AWM** ([abUX, ZC3a])
  We strengthened the explanation of why AWM is structurally sampler-agnostic and timestep-decoupled: its loss depends only on $x_0$ and forward-noised $x_t$, not on the particular reverse trajectory or intermediate states. To substantiate this, we **added new experiments** using both **first-order ODE and first-order SDE samplers on OCR and ImageReward**, showing consistent reward improvements and confirming that AWM’s gains are robust across different samplers.

* **Additional Experiments and Empirical Analyses** ([pVy6, abUX, ZC3a, FSZB])
  We augmented the experimental section with (i) **new ImageReward** results on SD3.5M, including a detailed breakdown of update-step efficiency, per-step cost, and total GPU-hour speedup; (ii) **new comparisons** against Dance-GRPO (on FLUX) and DiffusionNFT (on SD3.5M); (iii) an **additional $\beta = 0.3$ run** to probe training stability; (iv) qualitative checks on **diversity across random seeds** for the AWM-finetuned FLUX model; and (v) **evaluations under both first-order ODE and SDE samplers** on OCR and ImageReward, confirming that AWM’s gains are robust across different sampling schemes.



We hope our pointwise responses below could clarify all reviewers’ confusion and alleviate all concerns. We thank all reviewers’ effort and time again.

Authors

---

### Meta-Review · Area_Chair_CFTj · 2026-01-06

**Summary:**

This paper studies the RL based post training of diffusion models, and proposed an advantage weighted matching (AWM) algorithm which has the benefit of variance reduction. The contributions lie both in the theoretical aspect by linking DDPO and Noisy DSM, as well as empirically showing fast convergence compared to Flow-GRPO.

The paper received mixed reviews. On the one hand reviewers find the theory part intriguing and the demonstrated speedups impressive, on the other hand several questions regarding the novelty, presentation, rigor of the proof, comprehensiveness of the evaluation are raised. The authors tried very hard in the rebuttal, but after going through their responses, it doesn't seem like the concerns are fully addressed given the current shape of the draft. Given the high bar of ICLR, the AC believes that it's most appropriate to recommend reject but encourage the authors keep improving it.

**Reviewer Concerns:**

Lots of presentation/theory related concerns were clarified, but the points brought up by Reviewer ZC3a (regarding rigor of the claims) and Reviewer abUX (regarding weight collapse and other baselines) remain more to desire.

**Reviewer Scores:**

I see a potential for all reviewers to increase their rating after the rebuttal, but realistically the average rating would still fall short of the acceptance threshold.

---

### Decision · Program_Chairs · 2026-01-26

Reject